# Catalytically controlled formation of coumarin-based hydrogelator enables colorimetric ferrous ion detection in sol and hydrogel

Nikita Das[1], Samir Mandal[2], Sib Sankar Mal[3], Suryasarathi Bose[2] & Chandan Maity [1] ✉

In-situ generation of a hydrogelator from non-assembling precursors offers an effective strategy for preparing supramolecular hydrogel materials with precise spatiotemporal control. These hydrogels hold broad potential for applications ranging from theranostics to chemical sensing. Herein, we report a method for the in-situ formation of a coumarin-based supramolecular hydrogelator by simply mixing aqueous solutions of two non-assembling precursors under ambient conditions. The formation of the hydrogelator, its subsequent self-assembly into a hydrogel network, and the resulting material properties can all be modulated via acid catalysis. The hydrogelator exhibits excellent selectivity toward Fe(II) ions, providing a distinct colorimetric response with a linear correlation and a notable detection limit. Additionally, the hydrogel material can be easily applied to disposable paper strips, enabling convenient and portable detection of Fe(II) ions. This system demonstrates strong potential for addressing key challenges in Fe(II) ion sensing in both aqueous environments and self-assembled hydrogel states.

Supramolecular hydrogels are formed from low-molecular-weight (LMW) molecules, known as hydrogelators, that self-assemble via non-covalent interactions into a three-dimensional nanofibrous network capable of trapping and immobilizing water within its structure[1,2]. In recent years, a wide range of supramolecular hydrogel systems have been developed[3,4], with applications spanning petrochemicals[5], agricultural fertilizers[6], personal care products[7], therapeutics[8], and tissue engineering[9]. Supramolecular hydrogels are typically prepared by altering physical or chemical conditions, such as dissolving the hydrogelator at elevated temperatures, followed by cooling, applying ultrasound energy, or adjusting the pH of the solution[10]. However, achieving controlled formation and precise spatial distribution of these materials for smart applications remains a significant challenge. This difficulty primarily arises from the weak non-covalent interactions that drive the self-assembly of LMW molecules into supramolecular structures. To overcome these limitations, in-situ generation of hydrogelators from non-assembling building blocks offers a promising strategy for exerting spatiotemporal control over hydrogel formation[11]. The use of catalysts to drive covalent bond formation between precursor molecules enables the controlled synthesis of hydrogelators, providing enhanced control over the

hierarchical self-assembly process. This approach also improves the tunability of hydrogel properties and enhances their responsiveness to analytes in biological environments[12,13]. In this context, the in-situ generation of hydrazone-based LMW hydrogelators under ambient conditions, facilitated by acid or nucleophilic catalysts, represents an interesting method for the controlled fabrication of supramolecular hydrogels[14,15].

Iron (Fe) is the most abundant transition metal in living organisms that plays numerous vital biological roles, including electron transportation, gene regulation, macrophage polarization, and mitochondrial oxidative respiration[16,17]. The ferrous ion (Fe(II)) is particularly crucial for cellular redox haemostasis, specifically during oxidative phosphorylation in the mitochondria[18]. Beyond biological systems, Fe(II) ions also find widespread use in various industrial applications, such as colorants in the textile and cosmetic industries, and as fillers in different formulations[19]. However, elevated concentrations of this metal ion can be hazardous and toxic. Excessive levels of this metal ion have been linked to organ dysfunction, particularly affecting the liver, kidneys, and causing gastrointestinal complications such as bleeding and ulcerations[20]. As a result, the selective and sensitive detection of Fe(II) has attracted considerable global attention. A

[1](Organic)Materials and Engineering Laboratory, CNBT, Vellore Institute of Technology (VIT), Vellore, Tamil Nadu, 632014, India. [2]Department of Materials Engineering, Indian Institute of Science (IISc), Bangalore, 560012, India. [3]Materials and Catalysis Laboratory, Department of Chemistry, National Institute of Technology Karnataka (NITK), Surathkal, 575025, India. ✉e-mail: chandan.maity@vit.ac.in; chandanmaitylab@gmail.com

variety of spectroscopic techniques have been employed for Fe(II) detection, including absorption spectroscopy[21], electrochemical methods[22], fluorescence techniques[23], chemiluminescence[24], and colorimetry[25]. However, the development of effective probes often involves complex, multi-step syntheses, which can hinder the efficiency and practicality of Fe(II) detection. Furthermore, many of these methods require specialized instrumentation, making them less accessible in resource-limited settings. Hence, there is a growing demand for simple, cost-effective, and highly selective molecular probes that enable easy and direct detection of Fe(II) ions without the need for elaborate procedures or equipment.

There is significant interest in developing hydrogel-based platforms for sensing and therapeutic applications[26]. When immobilized in a test medium for analyte detection (particularly for metal ions), hydrogels allow analytes to readily diffuse through their porous network, thereby facilitating efficient detection[27]. However, preparing such hydrogel materials under ambient conditions is crucial to preserving the integrity and activity of the embedded sensing components. We envisioned that incorporating a chromophore into the hydrogelator structure that can be generated under mild, ambient conditions would provide effective metal-binding sites in aqueous environments, resulting in a visible colour change upon analyte interaction. Herein, we report the selective detection of Fe(II) ions in aqueous solution using a coumarin-based hydrogelator (C-HyG, Fig. 1). C-HyG is formed in situ by simply mixing aqueous solutions of two non-assembling building blocks under ambient conditions. Its formation and subsequent self-assembly into a supramolecular hydrogel can be finely tuned via acid catalysis. In aqueous media, C-HyG exhibits high selectivity toward Fe(II) ions over other commonly encountered metal cations, producing a distinct optical (colorimetric) response. Moreover, the hydrogel can be easily coated onto disposable paper strips through a simple impregnation process, enabling straightforward and portable detection of Fe(II) ions. This system shows strong potential to address current challenges in Fe(II) sensing, both in solution-based and hydrogel-based formats. The platform holds promise for real-world applications in biomedical diagnostics, environmental monitoring, and industrial process control.

## Results

Supramolecular hydrogels are excellent candidates for the fabrication of analyte sensor devices, primarily due to their high-water content and ability to retain water within a structured network. These materials can be conveniently prepared in situ by mixing non-assembling molecular building blocks, offering enhanced control over the hierarchical self-assembly process. This approach not only enables easy tunability of the hydrogel's physical properties but also enhances its responsiveness to analytes, particularly in biological environments. A general strategy for developing a molecular probe to sense Fe(II) ions under ambient and/or physiologically relevant conditions should consider the following criteria:

(a) The probe should be readily accessible, either commercially available or easily prepared by simple mixing under ambient conditions, and must be water-soluble to ensure compatibility with aqueous environments.

(b) It should contain a specific binding motif for Fe(II) ions (or other biologically relevant ions) that enables a straightforward readout, such as colorimetric detection[28], without the need for specialized instruments or equipment.

(c) The probe should be immobilizable on a solid support, such as a paper strip[29], to create a portable sensing device that allows for convenient, high-throughput detection, particularly useful in resource-limited settings.

Conventional methods for covalent fixation of molecular probes to polymers or glass substrates suffer from several limitations. These include a reduced binding and/or sensing efficiency of the probes compared to their performance in homogeneous solution, as well as the requirement for specific functional groups or attachment sites to enable successful fixation onto solid surfaces[30,31]. Furthermore, a major drawback of solid-state sensing devices is their inherently dry nature, which makes them less compatible with biological media or environmental samples, where hydration is critical. In contrast, hydrogel-based sensing platforms offer a hydrated environment that better supports analyte interaction. However, preparing hydrogel materials under ambient conditions for sensing applications remains challenging, yet essential to preserve the structural integrity and functional activity of embedded sensing components. This becomes especially important in point-of-care settings, where controlling environmental factors may be difficult. The ability to form hydrogels simply through mixing, without the need for harsh conditions or complex procedures, greatly simplifies the sensing process and enhances their practicality for real-world applications.

### In-situ hydrogelator and hydrogel preparation

In line with the design principles for developing molecular probes capable of detecting Fe(II) ions under ambient conditions, we have designed a coumarin-based hydrogelator (C-HyG, Fig. 2a). This molecule features a coumarin moiety linked to three hydrazone groups, and is flanked by a hydrophilic guanidine unit. Coumarin derivatives are well-known for their sensing capabilities toward anions, cations, and reactive species due to their stable optical signals, structural flexibility, and biocompatibility[32]. However, readily accessible coumarin-based probes specifically tailored for Fe(II) detection remain scarce[33,34]. Notably, C-HyG can be conveniently prepared in situ by simply mixing a solution of a coumarin aldehyde derivative (CA, Fig. 2a) with an aqueous solution of guanidine hydrazide (GH, Fig. 2a) in phosphate-buffered saline (PBS) at room temperature. The hydrogelator is formed via hydrazone bond formation, involving the reaction of one equivalent of GH with three equivalents of CA. The structure of the in-situ-

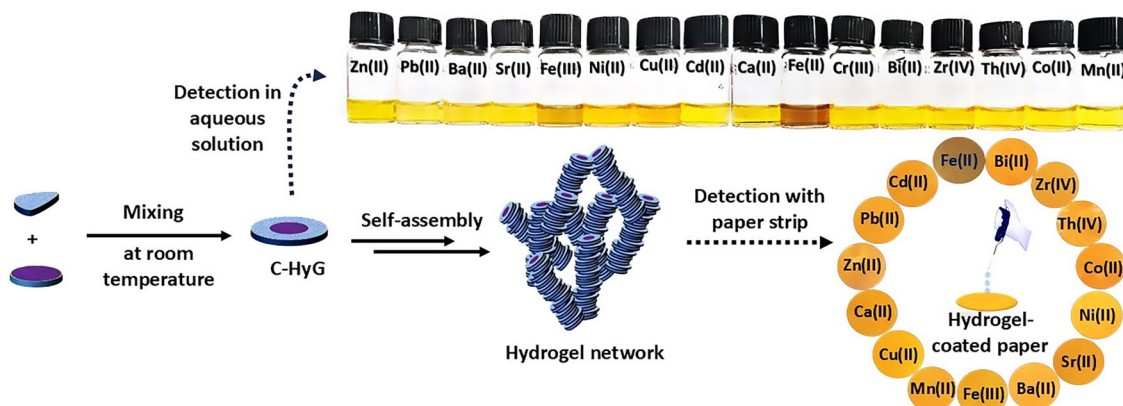

**Fig. 1 | Schematic of the coumarin-based hydrogelator (C-HyG) system.** A schematic description of in-situ formation and self-assembly of coumarin-based hydrogelator (C-HyG) and its application in selective Fe(II) detection in aqueous solution and on paper-based platforms.

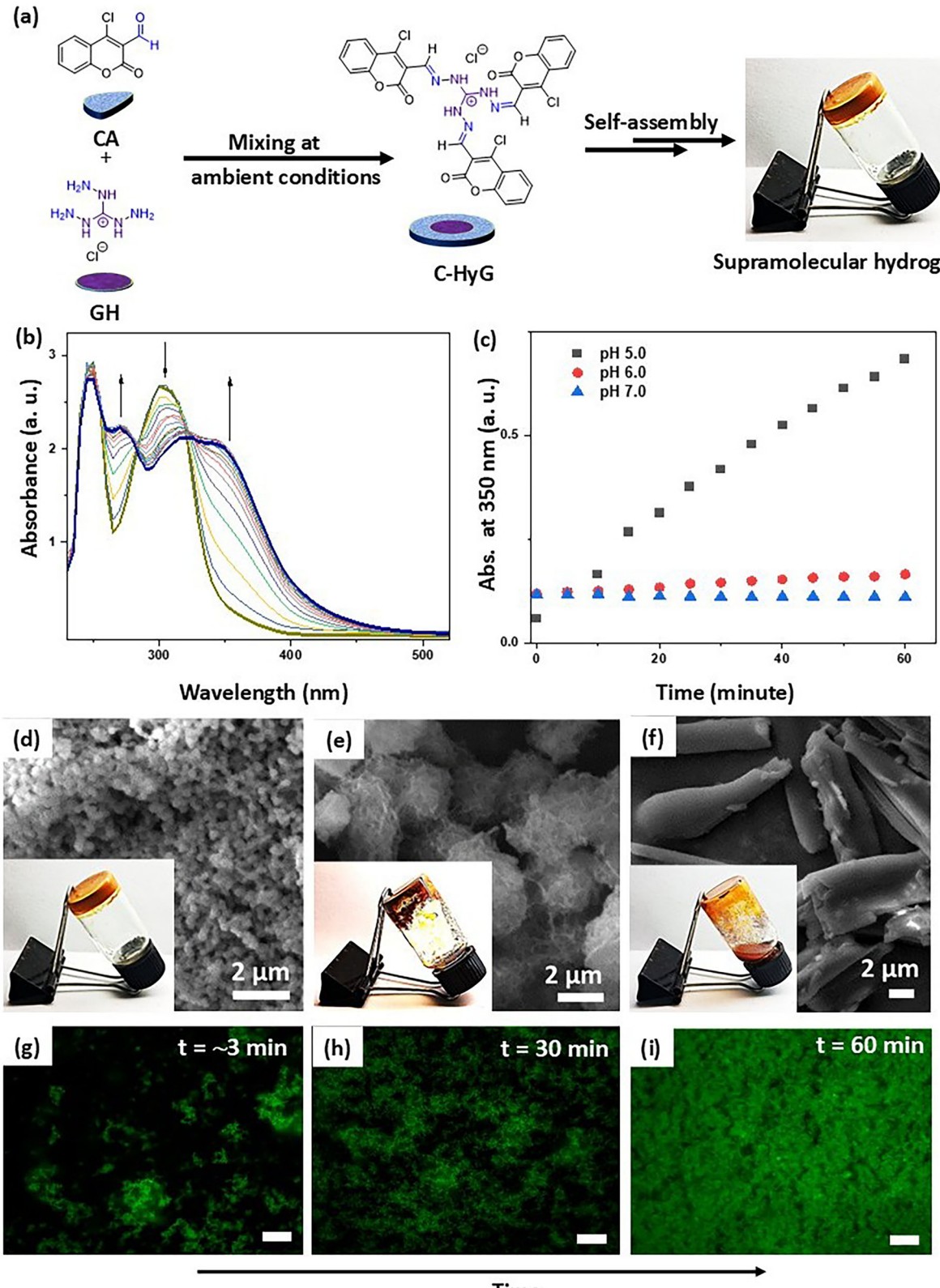

**Fig. 2 | In situ formation and structural characterization of hydrogel material under varying pH conditions. a** In-situ formation of C-HyG by mixing aqueous solution of CA and GH, leading to supramolecular hydrogelation at ambient condition. **b** UV–vis spectra showing hydrazone product formation in MeOH/PBS (pH 5.0) over 60 min (green line: 0 min; blue line: 60 min) after mixing CA (90.0 µM) and GH (30.0 µM). **c** Conversion to hydrazone product followed by UV–visible spectroscopy measured at 350 nm in pH 5.0 (black square), pH 6.0 (red) and pH 7.0 (blue). The conversion was followed after mixing CA (90.0 µM) and GH (30.0 µM) at ambient conditions. **d**–**f** Self-assembled material formation at different conditions (scale bar: 2 µm)—**d** SEM image of the material at pH = 5, **e** SEM image of the material at pH = 6, and **f** SEM image of the material at pH = 7; inset: photograph of obtained material; **g**–**i** Fluorescence micrographs for in-situ evolution of hydrogel network at pH 5, emission wavelength ($\lambda_{em}$) = 500–570 nm, scale bar: 10 µm.

generated C-HyG has been confirmed by comparison with the isolated compound, using $^1$H NMR spectroscopy, mass spectrometry, and comparative FTIR spectral analysis (Supplementary Figs. S1–S3).

Hydrazone functionalities act as effective hydrogen-bond donors and acceptors, facilitating the non-covalent interactions essential for self-assembly[35]. Interestingly, when aqueous solutions of GH (1 eq.) and CA (3 eq.) were mixed in a 1:1 PBS buffer at pH 5.0 with a cosolvent under ambient conditions, an orange-brown hydrogel was formed. Due to the limited aqueous solubility of CA, various water-miscible organic co-solvents were tested to optimize hydrogel formation (Supplementary Fig. S4). Hydrogels were successfully obtained within 10 min using dimethylformamide (DMF), tetrahydrofuran (THF), and dimethyl sulfoxide (DMSO), while methanol (MeOH) required approximately one hour for gelation (Supplementary Table S1). The minimum gelation concentration (MGC) was found to be 17.5 mM in THF and 22.0 mM in DMF, DMSO, and MeOH, as determined by vial inversion tests performed after allowing the mixtures to rest overnight (Supplementary Fig. S5, Tables S3 and S4). Below the MGC, precipitation was observed instead of gel formation. The relative abundance of C-HyG in aprotic polar solvents such as DMF and THF was ~88% as assessed by LC–MS analysis (Supplementary Table S2). In contrast, DMSO yielded a weak gel, predominantly containing di-functionalized hydrazide byproduct (bis-C-HyG, ~99% abundance).

This shift in product distribution is likely due to the higher density of DMSO, which restricts molecular mobility and favours the formation of dimeric structures (bis-C-HyG) over C-HyG. Rheological analysis supported these observations, showing that the gel formed with DMSO exhibited the lowest mechanical strength ($G'_{max}$ = 0.53 kPa, Supplementary Fig. S6), compared to those prepared in MeOH ($G'_{max}$ = 0.97 kPa), THF ($G'_{max}$ = 34.7 kPa), and DMF ($G'_{max}$ = 77.8 kPa). These results indicate that while di-C-HyG is capable of forming hydrogels, greater mechanical robustness is achieved when the proportion of C-HyG is higher relative to its mono- and di-substituted counterparts, as observed in gels formed using DMF and THF.

Next, the influence of an acidic environment (pH 5.0) on the formation of C-HyG and its subsequent hydrogelation was investigated. Compared to the hydrogel obtained in PBS buffer at pH 5.0, the same reaction mixture at pH 6.0 resulted in a weak gel that failed the vial inversion test, indicating insufficient mechanical integrity. At pH 7.0, mixing the CA and GH precursors produced only a viscous solution. This variation is attributed to the role of acid catalysis, which enhances the rate of hydrazone bond formation —a key step in C-HyG assembly and supramolecular gelation[14,36]. The formation of C-HyG was further monitored using UV–visible spectroscopy. The precursor CA showed a maximum absorbance at 305 nm, which gradually decreased upon the addition of GH. Simultaneously, two new peaks emerged at 205 nm and 350 nm, indicating the formation of C-HyG (Fig. 2b). The formation of the hydrazone bond was specifically followed by monitoring the increase in absorbance at 350 nm (Fig. 2c). The rate constant for hydrazone bond formation was determined by mixing GH (30.0 μM) in PBS buffer (pH 5.0, 6.0, or 7.0) with a methanolic solution of CA (90.0 μM), maintaining a consistent 1:1 (v/v) ratio of buffer to methanol in all experiments. Kinetic analysis, based on the increase in absorbance at 350 nm (corresponding to the absorption maximum of purified C-HyG), indicated second-order behaviour, confirming the gradual formation of the hydrazone bond over time. At pH 7.0, the rate of hydrazone bond formation was $0.2 \times 10^{-5}$ L mol$^{-1}$ min$^{-1}$ (Supplementary Fig. S7c, and Table S5). However, the rate increased to 185.5 times faster at pH = 5.0 (rate constant, $k = 37.1 \times 10^{-5}$ L mol$^{-1}$ min$^{-1}$, Supplementary Fig. S7a), and it was only 8.0 times faster at pH = 6.0 (rate constant, $k = 1.6 \times 10^{-5}$ L mol$^{-1}$ min$^{-1}$, Supplementary Fig. S7b). To evaluate product formation, LC–MS analysis of the hydrogel at pH 5.0 showed ~63% yield of C-HyG along with 29% of a bis-C-HyG (Supplementary Table S6). Notably, increasing the molar ratio of CA to GH enhanced gelation efficiency. Using 4 equivalents of CA yielded 84.4% C-HyG, while 5 equivalents increased the yield to 92.1% (Supplementary Table S7). This was accompanied by the improvement in the hydrogel's mechanical strength, with $G'_{max}$ rising from 2.7 kPa (with 3 eq.

CA) to 93.9 kPa (with 4 eq. CA), and 101.5 kPa (with 5 equivalents of CA) (Supplementary Fig. S8a). These results suggest that higher CA concentrations facilitate more efficient C-HyG formation, leading to a stronger and robust hydrogel network. Additionally, rheological tests confirmed the stability of C-HyG hydrogels, with $G' > G''$ up to 1% strain (Supplementary Fig. S8b). These findings demonstrate that both precursor concentration and pH can be tuned to control hydrogel formation and properties. Specifically, acid catalysis accelerates the in-situ generation of C-HyG, promoting an interconnecting hydrogel network formation[11]. Conversely, the slower rate of hydrogelator formation at higher pH resulted in a poorly structured network as a weak gel or viscous material. It is worth noting that C-HyG was purified to approximately 96.2% purity, as confirmed by LC–MS analysis. However, gelation using the purified compound via a solvent-switching method was unsuccessful. C-HyG displayed limited solubility in DMSO, requiring heating to reach the desired concentration. Upon addition of buffer solutions, no hydrogelation occurred; instead, a yellow-brown precipitate formed without further network development. These findings suggest that purified C-HyG alone is insufficient for gel formation, displaying the importance of in-situ generation of C-HyG and/or specific solvent environments in promoting supramolecular assembly.

Scanning electron microscope (SEM) analysis revealed that the material formed at pH = 5.0 exhibited spheroidal aggregates, which collectively arranged into a stratified architecture (Fig. 2d, and Supplementary Fig. S9a-c). Further analysis using field emission scanning electron microscopy (FESEM) enabled the visualization of an interconnecting nanofibrous network consisting of fibres with an average diameter of ~46 ± 10 nm (Supplementary Fig. S10), indicating that these nanofibrillar networks interconnect to an intricate web forming the spheroidal aggregates. A similar spheroidal morphology was also observed at pH 6.0. However, the aggregates were significantly larger, averaging ~1.8 μm in diameter (Fig. 2e, and Supplementary Fig. S9d–f). These enlarged aggregates compromised the structural integrity of the hydrogel, as a weak gel was obtained. In contrast, the material formed at pH 7.0 displayed rod-like structures, with lengths of ~13.0 μm and widths of ~4.0 μm (Fig. 2f and Supplementary Fig. S9g–i). The influence of H$^+$ ions on hydrazone bond formation is clearly critical in determining the resulting properties of the C-HyG hydrogel. It is worth noting that under strongly acidic conditions (pH < 3), the reaction proceeds slowly, likely due to protonation of the hydrazine group, which reduces its nucleophilicity and hinders hydrazone bond formation. At pH 5.0, the addition of a colourless aqueous solution of GH (30.0 mM in PBS buffer) to a pale-yellow methanolic solution of CA (90.0 mM) at room temperature led to the formation of an orange-coloured mixture, which gradually turned brown and formed a stable hydrogel (Supplementary Video V1).

Fluorescence microscopy was employed to visualize the hydrogel network without the use of any external fluorophores. Shortly after mixing CA and GH at pH 5.0, initial aggregates were observed (Fig. 2g), which progressively clustered and expanded to form an interconnected network (Fig. 2h). Over time, this network evolved into a dense, homogeneous fibrous structure characteristic of hydrogel formation (Fig. 2i, Supplementary Fig. S11a–c). The emergence of C-HyG aggregates corresponded to a gradual increase in fluorescence emission (Supplementary Fig. S11d, and Supplementary Video V2), further indicating the nucleation and growth of the hydrogel network. Additionally, confocal laser scanning microscopy (CLSM) confirmed the presence of a dense and continuous fibrous network in the material formed at pH 5.0 (Supplementary Fig. S12), supporting the formation of a robust supramolecular hydrogel.

Overall, efficient hydrogelation depends not only on reaction kinetics but also on the selective formation of hydrazone products. While acid catalysis promotes hydrazone bond formation, the generation of C-HyG is crucial for gelation. For example, in the presence of DMSO as co-solvent, bis-C-HyG dominates at pH 5.0 (~99%, Supplementary Table S2), yielding a weak and mechanically unstable hydrogel. In contrast, increasing the CA ratio at pH 5.0 shifts the product distribution toward >90% C-HyG (Supplementary Table S7), resulting in a robust and stable hydrogel.

## Metal ion sensing at the solution state

We next investigated the metal ion sensing capability of C-HyG in aqueous solution. The optical response of C-HyG was evaluated in the presence of various metal ions. No significant changes were observed upon exposure to a broad range of cations, including Ca(II), Zn(II), Ni(II), Cu(II), Co(II), Mn(II), Cd(II), Sr(II), Bi(II), Cr(III), Zr(IV), and Th(IV). These ions were selected to represent biologically abundant species such as Ca(II), and Mn(II), essential transition metals such as Zn(II), Cu(II), and Ni(II) that may compete for coordination, and environmentally relevant heavy metals such as Cd(II), Cr(III), and Th(IV) that could act as potential interferents in sensing applications. Interestingly, Fe(II) resulted in a strong and selective response, turning the colour from yellow to dark brown (Fig. 3a). Fe(III) also led to a slight colour change to pale brown, which could be seen by eye, but it was much less noticeable than the response to Fe(II). UV–vis spectroscopy was employed to monitor the spectral changes of C-HyG in the presence of metal ions (Fig. 3b). Measurements were conducted in HEPES buffer (50 μM) with 10% DMSO as a cosolvent to enhance the solubility of C-HyG at room temperature. Notably, purified C-HyG was used for the metal ion sensing studies. When C-HyG was prepared in situ and directly used for sensing, minor changes in the spectral response were observed compared to the use of the purified C-HyG. The C-HyG solution exhibited two prominent absorbance bands centred at 275 nm and 360 nm. No notable spectral changes were detected for the majority of tested cations. However, upon the addition of Fe(II), a red shift of approximately 15 nm and a substantial decrease in absorbance at 375 nm were observed (Fig. 3b), indicating a strong interaction between C-HyG and Fe(II). To further explore the nature of this interaction, the brown-coloured material was isolated and characterized by FTIR spectroscopy. Notable shifts were observed at 1686 cm$^{-1}$ (C=O stretching) and 3225 cm$^{-1}$ (N–H stretching), suggesting coordination of Fe(II) with the carbonyl oxygen of the coumarin moiety and the nitrogen atom of the hydrazone group (Supplementary Fig. S13). Additionally, X-ray photoelectron spectroscopy (XPS) revealed shifts of approximately 0.4 eV in the O 1s binding energy and 0.3 eV in the N 1s binding energy upon coordination with Fe(II), indicating the involvement of the coumarin carbonyl oxygen and hydrazone nitrogen atoms in metal binding (Supplementary Fig. S14a–d)[37]. In contrast, no significant shift was observed in the C 1s region, suggesting minimal perturbation of the carbon environment. Furthermore, the XPS spectrum displayed a characteristic Fe $2p_{3/2}$ peak ~709 eV and a corresponding $2p_{1/2}$ peak around 721 eV, characteristic of Fe(II)[38]. Notably, no signals attributable to Fe(III) were observed, confirming that iron is present in the +2 oxidation state (Supplementary Fig. S14e).

Further insights into the binding mechanism were obtained via $^1$H NMR titration in DMSO-d$_6$. The hydrazone proton signal decreased in intensity and exhibited a slight downfield shift with increasing concentrations of Fe(II). Additionally, peak broadening indicated the formation and possible separation of the C-HyG-Fe(II) complex from the solution, suggesting the involvement of the –C=N– moiety in coordination (Supplementary Fig. S15). This is presumably due to coordination of the hydrazone nitrogen atoms with the metal ions, whereas the NH group does not participate, likely due to its less favourable electronic configuration for metal binding[39–41]. Besides, a broadening of the aromatic proton signals was observed for Fe(III) ions, consistent with the paramagnetic nature of Fe(III)[42].

Next, the stoichiometry of the complex formed between Fe(II) ions and C-HyG was determined using Job plot analysis[43,44], which revealed a 1:1 binding ratio (Fe(II):C-HyG) (Supplementary Fig. S16). Based on this result, we propose a plausible complexation model wherein Fe(II) coordinates with C-HyG through nitrogen atoms of the hydrazone linkages and oxygen atoms from the coumarin units (Supplementary Fig. S17a, b, and Supplementary Video V3). This stoichiometry was further supported by high-resolution mass spectrometry, which showed a dominant peak at *m/z* 763, consistent with the estimated mass of the 1:1 Fe(II)–C-HyG complex (Supplementary Fig. S18). To further examine the interaction between C-HyG and Fe(II), a UV–vis titration experiment was conducted by gradually adding Fe(II) ions (0–125 μM) to a solution of C-HyG (50 μM)

(Fig. 3c). The absorbance at 375 nm increased with Fe(II) concentration and reached saturation at ~50 μM. The observed linear relationship with the Fe(II) concentration in the range of 0–50 μM indicates that C-HyG can be potentially used for quantitatively detecting Fe(II) ions. The limit of detection (LOD)[45] was calculated to be 32.1 μM (Supplementary Fig. S19). To evaluate the selectivity of C-HyG, competitive binding experiments were carried out by introducing other metal ions along with the Fe(II) ion. No significant interference was observed (Fig. 3d), confirming the high selectivity of C-HyG for Fe(II). It is important to note that C-HyG did not exhibit any distinguishable changes in its fluorescence spectra ($\lambda_{ex}$ = 375 nm) in the presence of metal ions. Besides, the response of C-HyG toward Fe(II) was examined under varying pH conditions. A visible colour change was observed between pH 4.0 and 8.0, indicating that C-HyG is suitable for Fe(II) detection under physiologically relevant conditions. At highly acidic pH (<4.0), precipitation occurred, while a reddish coloration appeared at alkaline pH (>8.0) (Supplementary Fig. S20).

To evaluate the practical applicability of C-HyG for real-time Fe(II) detection, its performance was tested in Fe(II)-spiked aqueous samples prepared using potable water sources, including mineral water, tap water, river water and HEPES buffer solution[46,47]. UV–vis absorption spectroscopy was used to monitor changes in absorbance at 375 nm upon the addition of Fe(II) ions at concentrations ranging from 5 μM to 40 μM. In the presence of aqueous C-HyG (50.0 μM), a gradual decrease in absorbance was observed with increasing Fe(II) concentration (Fig. 3e) for water collected from all potable sources. This trend closely resembled the results obtained under controlled buffered conditions, confirming the sensitivity of C-HyG to Fe(II) even in complex water matrices. These findings demonstrate the potential of C-HyG for Fe(II) detection in real-world water samples, with consistent sensing behaviour in the sol state.

## Metal ion sensing in the hydrogel state

A hydrogel coating on an appropriate surface can serve as a rapid and efficient platform for detecting Fe(II) ions. To explore this, we investigated the use of the C-HyG-based hydrogel for fabricating a sensor device. Hydrogel-based sensors have shown advantages over solution-based systems, particularly due to the nanostructured gel surface, which enhances analyte interaction and signal response[48,49]. Utilizing C-HyG as both the hydrogelator and sensing element, we evaluated its feasibility for Fe(II) detection. Therefore, CA and GH precursors were mixed and applied onto filter paper, where in-situ hydrogelation produced disposable, ready-to-use paper-based sensors. Reflectance measurements of the hydrogel-coated paper strips showed approximately 69% reflectance (Fig. 3f). Upon exposure to aqueous solutions of various metal salts, no significant colour change or variation in reflectance was observed (Fig. 3f, g). However, when Fe(II) was applied to the orange paper strip, it turned dark brown, with reflectance dropping to about 30%. Furthermore, reflectance at 700 nm decreased progressively from 80% to 9% as the Fe(II) concentration increased from 0 to 37.5 mM (Supplementary Fig. S21). At higher concentrations, no further significant changes in the reflectance spectrum were detected. This approach offers a more straightforward and practical alternative to traditional probe-doping techniques[50] for sensor development.

## Discussion

In conclusion, we report an in situ method for preparing C-HyG via simply mixing aqueous solutions of two components, GH and CA. The rate of C-HyG formation can be modulated through acid catalysis, enabling hydrogelation under ambient conditions from readily available building blocks. We demonstrated the utility of C-HyG for the selective detection of Fe(II) ions in aqueous buffer, with a LoD of ~32 μM. Moreover, application of in-situ prepared C-HyG onto paper substrates produced a visible colour change in the presence of Fe(II), facilitating a portable and user-friendly sensing platform. This supramolecular system, featuring a convenient preparation method, rapid response, distinct optical changes, and ease of deployment, offers a promising and underexplored approach for designing metal ion sensors. We believe this in-situ hydrogel-based sensing strategy

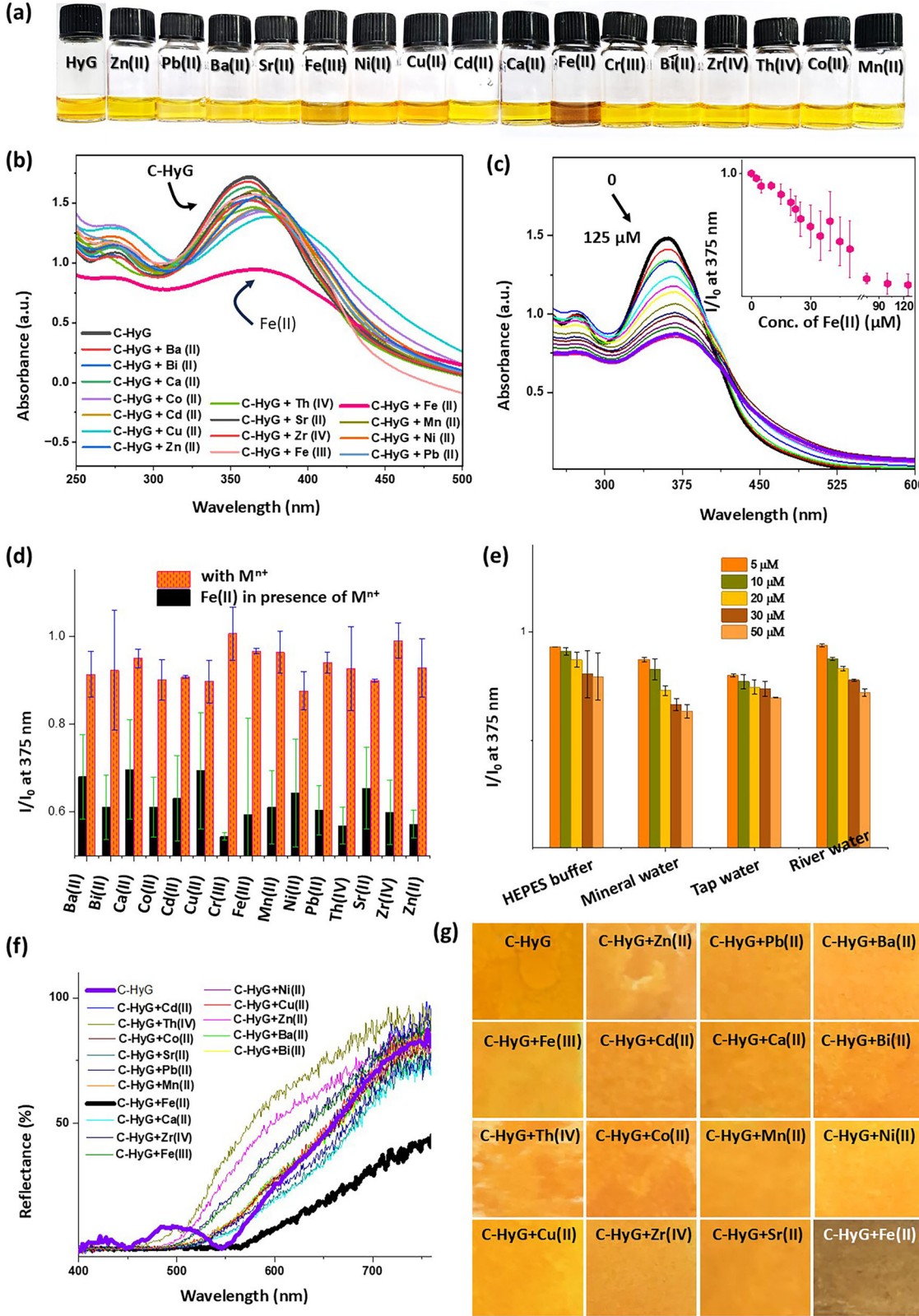

**Fig. 3 | Evaluation of C-HyG for Fe(II) detection in sol and gel state. a** Visual colour changes in C-HyG solution (50 µM, 9:1 water/DMSO) after addition of various metal ions. **b** UV–vis spectra of C-HyG (50 µM) with different metal cations. **c** Absorbance spectra of C-HyG with increasing Fe(II) concentrations (0–125 µM), inset: absorbance at 375 nm vs. Fe(II). **d** Competitive binding assay showing Fe(II) selectivity in the presence of other cations. **e** Real-time Fe(II) detection in HEPES buffer, mineral water, tap water, and river water using C-HyG. **f** Reflectance UV–vis spectra of hydrogel-coated paper strips treated with metal ions. **g** Photographs of paper strips after exposure to different metal salt solutions. Error bars represent standard deviations from three independent measurements.

provides a viable alternative to biomolecule-dependent assays (e.g., enzyme- or protein-based) in potential theranostic applications.

## Methods

### Gel formation

The same protocol was followed for all gelation experiments, including the determination of minimum gelation concentration (MGC), pH-dependent hydrogel formation, and the selection of a suitable cosolvent. The two building blocks, CA and GH, were dissolved separately. CA was dissolved in an organic solvent (due to its water insolubility), and GH was dissolved in buffer. Phosphate-buffered saline (PBS, 0.1 M) was used at three different pH values: 5.0, 6.0, and 7.0. The mixtures were prepared using a 1:1 ratio of buffer to co-solvent, with varying concentrations of CA and GH. Gelation was confirmed by inverting the vials after repeated intervals of time. For all characterisation techniques that required hydrogel ageing, the gels were allowed to rest overnight and then processed further as required.

### NMR

C-HyG (in purified powder form) was dissolved in DMSO-$d_6$, and the $^1$H NMR spectrum was recorded using a Bruker FT-NMR 400 MHz instrument. Chemical shifts ($\delta$) are reported relative to the residual solvent peak.

### High resolution mass spectroscopy (HR–MS) or liquid chromatography mass spectroscopy (LC–MS)

High-resolution mass spectrometry (HRMS) of purified C-HyG was performed using an XEVO-G2-XS-QTOF instrument in positive ESI mode. To determine the composition of gels formed under different pH and solvent conditions, LC–MS analysis was also carried out using the same instrument. For analysis, hydrogels were left undisturbed overnight, then dissolved in acetonitrile to prepare 1 ppm solutions, which were subsequently analysed. To evaluate co-solvent efficacy, 30 mM GH and 90 mM CA were mixed in a 1:1 ratio of pH 5.0 PBS buffer and the respective co-solvent. For experiments involving varying precursor compositions, the concentration of GH was fixed at 30 mM (pH 5.0), while the concentration of CA was varied from 90 to 150 mM.

### FT-IR spectroscopy

Gels prepared using a co-solvent in PBS buffer (0.1 M) were left undisturbed overnight at ambient temperature to stabilize. The stabilized gels were then lyophilized using a lyophilizer (Alpha 1–2 LD plus, Martin Christ) to obtain powdered samples. The powdered samples were analysed using a Shimadzu FT-IR spectrophotometer (Model: IR Affinity-1) in ATR mode. Spectra were recorded in absorbance mode over the range of 4000 cm$^{-1}$–400 cm$^{-1}$ with a resolution of 2 cm$^{-1}$, and averaged over 32 scans.

### Scanning electron microscopy (SEM)

Materials prepared in PBS buffer at different pH values (5.0, 6.0, and 7.0) were lyophilized, mounted on carbon stubs, and coated with gold using an SC7620 Mini Sputter Coater/Glow Discharge System. The coated samples were then imaged using a ZEISS EVO 18 scanning electron microscope (SEM). Images were captured in secondary electron (SE) mode at magnifications of 1000×, 5000×, 10,000×, and 15,000×, using an accelerating voltage of 10 kV.

### Field emission scanning electron microscopy (FESEM)

To analyse the fibrous morphology, a thin film of the hydrogel was formed by mixing two precursors (30 mM GH in PBS buffer (pH = 5.0), and 90 mM CA in methanol) and depositing the mixture onto aluminium foil. After the hydrogel material was formed in an hour, the foil was vacuum-dried, mounted onto a carbon stub, sputter-coated with gold using a Quorum QT 150S Plus, and examined using a Thermo Fischer FEI QUANTA 250 FEG.

### Fluorescence microscopy

Immediately after mixing the two precursors (GH and CA in a 1:3 ratio; 30 mM of GH and 90 mM of CA) in a vial, a 30 μL aliquot was dropped onto a glass slide and observed under a Leica microscope. The images captured at this point were labelled as time ($t$) = ~3 min. Thereafter, images were taken at time intervals.

### Confocal laser scanning microscopy (CLSM)

The precursors GH and CA were mixed in a 1:3 ratio (30 mM of GH and 90 mM of CA) in a vial, and immediately transferred to an imaging chamber. The mixture was allowed to rest for one hour to facilitate hydrogelation and was then observed using a Fluoview FV3000 (Olympus) microscope. A 488 nm laser was used to excite the system.

### Rheological analysis

All rheological experiments were performed using an oscillatory rheometer (TA Instruments DHR-3 rheometer). A parallel plate geometry with a 20 mm diameter steel plate was used for all tests, with the gap height fixed at 0.5 mm. To determine the viscoelastic region of the hydrogel, a freshly prepared hydrogel was placed on the lower plate, and a strain sweep was conducted in the range of 0.1–300% at a fixed angular frequency of 0.5 rad/s, whereas a frequency sweep was performed in the range of 0.1–100 rad/s under an iso-strain condition of 0.1%.

### X-ray photoelectron spectroscopy (XPS)

A 20.0 mM solution of purified C-HyG (in DMSO) was mixed with 2 equivalents of Fe(II) to obtain a precipitate, which was filtered, dried, and analysed using X-ray Photoelectron Spectroscopy (ULVAC-PHI VersaProbe 4) in elemental scan mode. A blank run of C-HyG was also recorded for comparative analysis.

### pH measurements

pH of all solutions or mixtures was measured using a mobile Aceteq pH meter (Model: PH-035).

### UV–visible absorbance studies for metal ion sensing

A stock solution of purified C-HyG (10.0 mM in DMSO) was used for colorimetric sensing experiments (for solution state sensing) of metal ions. For evaluating the colorimetric sensing activity, 15 μL of the C-HyG stock solution was added to 285 μL of DMSO (10% DMSO in aqueous buffer, v/v) in a quartz cuvette (1 cm path length), followed by dilution with 2700 μL of HEPES buffer to obtain a final volume of 3.0 mL. This resulted in a final C-HyG concentration of 50 μM. Subsequently, 2 equivalents of metal salt were added, and the absorbance spectra were recorded under ambient conditions. Metal ion stock solutions (10.0 mM) were prepared in water for UV–vis spectroscopic analysis. PBS buffer was avoided as it produced turbid solutions with certain metal ions.

### Stoichiometric analysis

For obtaining the Job's plot, the concentrations of C-HyG (in the sol state) and Fe(II) ions were varied while maintaining a constant total molar concentration of 200 μM. The relative absorbance was plotted on the y-axis against the molar fraction of Fe(II) on the x-axis. In all cases, C-HyG was initially dissolved in DMSO and then diluted with PBS buffer to maintain a 10% DMSO content in the buffer solution. The same solvent ratio was used for determining the limit of detection (LoD) and assessing the pH stability of C-HyG in the sol state. For pH-dependent analysis, PBS buffer solutions of varying pH (1–11) and a fixed ionic strength of 0.1 M were employed.

### Real-time sensing performance of C-HyG in the sol state for water samples

Three types of water samples were used for the analysis: (a) mineral water, obtained directly from commercially available brands, (b) tap water, collected from the common laboratory tap of the Department of Chemistry, Vellore Institute of Technology, Vellore, and (c) river water, obtained from the Palar River in Vellore, Tamil Nadu. Mineral and tap water samples were directly spiked with Fe(II) ions. In the case of river water, the sample was first decanted and filtered through a 0.2 μm syringe filter before being spiked

with varying concentrations of Fe(II) ions. Subsequently, 50 μL of the tris-C-HyG solution was added to each sample, and the absorbance was measured at 375 nm.

### Real-time detection of analytes using paper strip-based sensors
Test strips were prepared by applying the mixture of CA and GH on Whatman filter paper, which allowed in-situ formation of C-HyG hydrogel (30 mM) on the paper. A few drops of analyte solutions were then applied to the dried strips, followed by recording the colour change of the paper strips. Reflectance spectra for each strip (treated with either different metal ions or varying concentrations of Fe(II) ions) were measured in the range of 400–800 nm using a JASCO V-780 UV–vis spectrophotometer.

### Data availability
The authors declare that the data supporting the findings of this study are available within the main article, its Supplementary information, Additional Supplementary files, and Supplementary Data 1. Furthermore, raw spectroscopic data are available on Figshare at https://doi.org/10.6084/m9.figshare.30277072.

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

## Acknowledgements
C. M. acknowledges the financial support from VIT as a Seed grant (No. SG20240013). N.D. thanks DST for the INSPIRE fellowship (No. DST/INSPIRE/03/2022/000179). The authors also acknowledge the instrumental facility of VIT Vellore for carrying out this work.

## Author contributions
C.M. conceived the research. N.D. carried out the experiments. S.M. and S.B. performed the rheology experiments, and C.M. directed the research. C.M. and N.D. wrote the manuscript. S.S.M. provided suggestions on the experiments and revised the paper. All authors commented on the work and the paper.

## Funding

## Competing interests
A part of this work has been filed as an Indian patent application (Application No. IN202541037660) with C.M. and N.D. as inventors. The remaining authors (S.M., S.S.M., and S.B.) declare no competing interests.
