## [Transparent Peer Review file · Communications Chemistry]

Catalytically Controlled Formation of Coumarin-based Hydrogelator Enables Colorimetric Ferrous Ion detection in Sol and Hydrogel

Corresponding Author: Dr Chandan Maity

Version 0:

Reviewer comments:

Reviewer #1

(Remarks to the Author)

In this manuscript the authors present an approach to generate a coumarin-based hydrogelator (C-HyG) in situ via hydrazone condensation of a 4-formylcoumarin aldehyde (CA) and a tris-hydrazide linker (GH). The dynamic covalent gelator self-assembles under mild aqueous conditions (pH 5–6) into a supramolecular network that exhibits both colorimetric and fluorescent responses toward Fe(II) in solution, in bulk gel form, and as a paper-strip assay. I recommend that this manuscript undergo major revision to address the points detailed above.

- LC–MS of the freeze-dried hydrogel at pH 5 indicates only 60.3 % C-HyG, with 35.1 % di-hydrazide and 4.6 % side-products (Table S3) . The authors should isolate and fully purify C-HyG and determine its minimum gelation concentration alone to confirm that network formation arises exclusively from the intended gelator.
- Gelation in buffers containing 10 % and 20 % organic solvents is explored (Figure S10), but without analysis of the resulting gels. Characterizing the molecular composition of gels formed with each solvent fraction would reveal whether co-solvent content impacts product distribution or network purity.
- The description of how gels are prepared for SEM, rheology, FT-IR and other analyses lacks precise, consistent detail. In particular the solvent composition, pH, temperature, aging time and physical state (wet vs. dried, bulk gel vs. strip extract) used for each technique are not clearly stated. This makes it impossible to know whether observed differences arise from true material behavior or simply from mismatched experimental conditions. The authors should explicitly report, for every characterization method, the exact preparation protocol and environmental parameters so that the data can be interpreted and reproduced reliably.
- Morphological characterization relies solely on SEM of dried gels (Figure S6) . Inclusion of TEM samples would confirm fiber diameters, internal packing, and rule out drying artifacts.
- The authors are encouraged to quantify Fe(II) capture on the paper-strip sensors by measuring the decrease in Fe(II) concentration of the dipping solution before and after exposure, enabling calculation of $\mu\text{mol Fe}^{2+}$ per cm^2 of strip under identical sensing conditions.
- The manuscript does not report any ESI-MS data on the metal–ligand assemblies. Including ESI-MS and MS/MS fragmentation of extracts from both solution and gel samples would confirm the stoichiometry of Fe(II) coordination
- The manuscript would benefit from experiments to elucidate the coordination environment of Fe(II) within the C-HyG network and to explain its selectivity over other metal ions. A simple such as X-ray photoelectron spectroscopy, Mössbauer spectroscopy, or X-ray absorption spectroscopy could be applied to reveal ligand identity, oxidation state, and binding geometry and thereby substantiate the preferential binding of Fe(II).
- The experiments exploring gel formation in different organic cosolvent ratios (Figure S10) and the wide panel of metal salt tests (Figure 3 d and ESI Figure S14) read as descriptive data collections without a clear link to the manuscript main hypotheses . The authors should explain why each cosolvent fraction was chosen, for example whether ten percent versus twenty percent DMSO reflects particular application environments or probes specific solvent gelator interactions, and clarify how those results advance understanding of the self assembly mechanism. In the same way the rationale behind selecting each metal ion in the selectivity assay should be stated, for instance in terms of biological relevance or potential industrial interferences, and the results discussed to show how they support the claim of preferential Fe(II) sensing. Adding this context will convert isolated observations into a cohesive narrative that reinforces the study scientific impact.

There are also minor comments about the figures and numerous typos present in the manuscript:

- SEM micrographs in panels d–f require uniform, high-contrast scale bars and a slightly larger inset photograph of the vial or repositioning of the inset outside the micrograph.
- Fluorescence images in panels g–i should include scale bars and explicit labels of the time points (for example $t = 0$ min, t

- = 30 min, t = 60 min) directly on each micrograph to guide the reader through the evolution of the network.
- Titration plots in panels c–e should include error bars or shaded confidence intervals and clearly distinguish mineral water versus tap water curves using different marker shapes.
 - Line 9–10: "its self-assembly to fibrous hydrogel material" should be replaced with "its self-assembly into fibrous hydrogel material"
 - Line 17–18: "hydrogelators, which can self-assembly through non-covalent interactions" should be replaced with "hydrogelators, which can self-assemble through non-covalent interactions"
 - Line 19–20: "numerous supramolecular hydrogel systems has been reported" should be replaced with "numerous supramolecular hydrogel systems have been reported"
 - Line 26: "Despite these challenges,spatiotemporal control" space is missing
 - Line 36: "prepared at ambient conditions form easily available building blocks" should be replaced with "prepared at ambient conditions from easily available building blocks"
 - Line 38–39: "particularly during" should be replaced with "particularly during"
 - Line 43: "As a results, selective and sensitive detection" should be replaced with "As a result, selective and sensitive detection"
 - Line 49: "high selectivity, and straightforward readout method" should be replaced with "high selectivity, and straightforward readout method"
 - Line 75: "analyte sensor devices because it contains" should be replaced with "analyte sensor devices because it contains"
 - Line 77: "control in the hierarchal assembling process" should be replaced with "control in the hierarchical assembling process"
 - Line 114–116: "could not supports its own weight" should be replaced with "could not support its own weight"
 - Line 116–118: "at pH 7.0,a viscous material were produced." Space is missing
 - Line 151: "precipitation occured below CGC." should be replaced with "precipitation occurred below CGC."
 - Line 153–154: "resulting poor nucleophilicity." should be replaced with "resulting in poor nucleophilicity."
 - Line 166–167: "oscillatory rheology was employed The reaction" dot is missing
 - Line 171–173: "in 2 hours.Notably, a hydrogel" space is missing
 - Line 178: "Notable, there was no pronounced change observed" should be replaced with "Notably, there was no pronounced change observed"
 - Line 180–181: "changed form yellow to pale brown" should be replaced with "changed from yellow to pale brown"
 - Line 252–253: "The hydrogeltor C-HyG exhibits excellent sensitivity" should be replaced with "The hydrogelator C-HyG exhibits excellent sensitivity"
 - Line 255: "added advantage of its portability introduces" should be replaced with "added advantage of its portability introduces"

Reviewer #2

(Remarks to the Author)

Dear author(s),

I have some questions and comments.

1. What do you mean by immobilizing water? Could you explain very briefly in the text?
2. Please cite after mentioning the application to avoid block references (Lines 19-21).
3. Have you tested detection in a real aqueous or biological environment where sample treatment is essential? In the manuscript the only thing you mention is use in mineral water and tap water as a medium.
4. I think research should focus on the synthesis and characterization of coumarin-based hydrogelators.

What you mention is that "We believe that this in situ prepared hydrogel system could be beneficial for the development of readily accessible chemosensor alternatives to biomolecule-based assays (such as those using enzymes or proteins) in theranostic applications." This is not a ideal environment as you assume. In addition, what are the detection limits and linearity?

In Figure 3, I observe that UV-Vis detection deviates from the proposed selectivity in the presence of other divalent cations. Specificity is not observable.

Reviewer #3

(Remarks to the Author)

The manuscript presents an in-situ formation of a hydrogelator and its hydrogelation due to the formation of hydrazone and the application of such a system in Fe²⁺ detection. The authors use the design of hydrazone formation between aldehyde and hydrazine, which has been shown in the past to form hydrogels of different types. However, they functionalized with a new coumarin derivative for colorimetric detection of iron salt in solution and on a paper-strip based detection platform. The work demonstrates some interesting observation on the hydrogelation properties, including pH dependence and acid-

catalyzed process and selective iron salt detection capability. However, there are several claims and observations in the manuscript which are not supported by the data, as detailed below. Also, some of the experiments lack the details needed to fully appreciate the work. Therefore, I recommend a major revision.

1) " Interestingly, when mixing aqueous solutions of CA and GH in a 1:1 ratio of hydrazide to aldehyde functional groups at pH = 5.0, under " vs " mixing precursor solutions of guanidine 81 hydrazide (GH) and coumarin aldehyde (CA) taken in 1:3 ratio in".

I imagine that the precursors were mixed in 1:3 ratio but in the main text and SI has different ratios written which are not consistent.

2) " Additionally, when the solution of CA and GH were mixed at pH 7.0, a viscous material were produced. This change can be ascribed to the increased rate of hydrazone bond formation of C-HyG due to acid catalysis, which influences the formation of supramolecular material".

This claim is not fully supported. Based on the data in Table S3, there are different ratios of product at different pH. Thus, kinetics cannot be the only factor, the chemical compositions will also contribute. Authors should consider doing the reaction in excess of CA to ensure to complete the reaction with triple substitution and then may be comparison can be made easily. In any case, authors should clarify between the chemical composition of the hydrogel vs the kinetics to influence the gel property.

3)" These results suggested that acid catalysis enhanced the rate of hydrogelator formation, facilitating hydrogelation by creating an interconnecting hydrogel network." Again here, why it cannot be due to the relative ratio of different product. Table S3 is not very intuitive, why the conversion of triple hydrazone product is higher in pH 7 compared to pH 5. I think, author needs to also comment on the stability of the hydrogel based on the chemical composition in different pH and not just kinetics.

4)Fig. 2 b,c, why the absorbance is in a. u.?

5)" (e-g) Fluorescence micrographs for in situ evolution of hydrogel network at pH 5 ($\lambda_{ex} = 490 \text{ nm}$)." e-g are not fluorescence micrographs. Also, they should provide the time details of the fluorescence micrographs. At what time these images were taken?

6)Figure 2f does not look needle-like. It seems to be rolled up structure. More images can be provided in the SI.

7)" showed a general broadening of peaks, indicating that C-HyG is more specific to Fe(II) as compared to Fe(III)" This fact is not very obvious from the data in Fig. S13

8)"which indicated a binding stoichiometry of 2:3 for the (Fe(II):C-HyG) complex" Here we do not know the actual concentration of the hydrazone. I imagine the concentration comes following the table S3 which shows the presence of multiple species. Thus, the 2:3 binding ratio seems like not accurate. Further, author should show if the % of various product change upon binding with FeII.

9)In the SI, the details of gelation experiment and the IR measurement needs to be clearer.

10)Fig. S5: Is this hydrogelation kinetics or kinetics for the formation of hydrazone? How was this obtained, from the UV-Vis or HPLC? Does it correspond to the formation of any hydrazone or triply functionalized derivative?

11)Typographical error "that can be prepared at ambient conditions form easily available building blocks".

Version 1:

Reviewer comments:

Reviewer #1

(Remarks to the Author)

The authors have addressed the reviewers' comments thoroughly. The manuscript is technically sound, clearly presented, and suitable for publication.

Reviewer #2

(Remarks to the Author)

Dear Authors:

I appreciate your consideration of the suggestions for improvements so that your manuscript can be published.

I have read everything carefully, so I inform you that the authors of reference 15 specify a supramolecular gelling agent at an oil/water interface to produce nanofibers, not hydrazones. Could you correct this? The other references are correct.

Reviewer #3

(Remarks to the Author)

The authors have revised the manuscript according to my comments. Therefore, I recommend its publication now.

Revision Report for Manuscript: COMMSCHEM-25-0362

Authors response to Reviewer/Editorial office comments:

We sincerely thank the reviewer for their time, thoughtful evaluation, and constructive suggestions, which have significantly contributed to improving the quality of our manuscript. In the revised submission, each comment has been carefully addressed. Reviewer comments are **underlined** for clarity, our detailed responses are provided in **blue**, and the corresponding changes in the manuscript are highlighted in **purple**.

Reviewer 1:

In this manuscript the authors present an approach to generate a coumarin-based hydrogelator (C-HyG) in situ via hydrazone condensation of a 4-formylcoumarin aldehyde (CA) and a tris-hydrazide linker (GH). The dynamic covalent gelator self-assembles under mild aqueous conditions (pH 5–6) into a supramolecular network that exhibits both colorimetric and fluorescent responses toward Fe(II) in solution, in bulk gel form, and as a paper-strip assay. I recommend that this manuscript undergo major revision to address the points detailed above.

Response: We sincerely thank the reviewer for their thorough evaluation and constructive feedback, which have significantly contributed to improving the quality and clarity of our manuscript. In the revised version, we have addressed the reviewer's concerns by incorporating additional experimental data, clarifying mechanistic insights, enhancing the discussion, and implementing the necessary changes in the revised manuscript.

1) LC–MS of the freeze-dried hydrogel at pH 5 indicates only 60.3 % C-HyG, with 35.1 % di-hydrazide and 4.6 % side-products (Table S3). The authors should isolate and fully purify C-HyG and determine its minimum gelation concentration alone to confirm that network formation arises exclusively from the intended gelator.

Response: We thank the reviewer for the thoughtful suggestion. We would like to mention that C-HyG was purified to approximately 96.2% purity through multiple methanol washing steps. A minor fraction (3.8%) was identified as the bis-substituted species, bis-C-HyG, based on LC-MS analysis. However, attempts to induce gelation using the purified C-HyG via a solvent-switching method were unsuccessful. Due to its partial solubility in DMSO, the required stock concentration could only be achieved upon heating. Upon addition of the solution to buffer media (pH 5.0, 6.0, or 7.0), a yellow-brown precipitate formed

immediately, with no gelation observed even after prolonged incubation. The solvent-switching procedure is briefly outlined as follows: purified C-HyG (30–50 mM) was dissolved in DMSO by heating and then added to aqueous buffer solutions at varying pH values. The resulting precipitation suggests that the purified C-HyG alone is insufficient to drive hydrogel formation, emphasizing the role of specific solvent environments likely in promoting supramolecular network assembly.

The following information has now been incorporated into the revised manuscript.

“It is worth noting that C-HyG was purified to approximately 96.2% purity, as confirmed by LC-MS analysis. However, gelation using the purified compound via a solvent-switching method was unsuccessful. C-HyG displayed limited solubility in DMSO, requiring heating to reach the desired concentration. Upon addition of buffer solutions, no hydrogelation occurred; instead, a yellow-brown precipitate formed without further network development. These findings suggest that purified C-HyG alone is insufficient for gel formation, displaying the importance of *in situ* generation of C-HyG and/or specific solvent environments in promoting supramolecular assembly.”

2) Gelation in buffers containing 10 % and 20 % organic solvents is explored (Figure S10), but without analysis of the resulting gels. Characterizing the molecular composition of gels formed with each solvent fraction would reveal whether co-solvent content impacts product distribution or network purity.

Response: For all gelation experiments, 50% co-solvent (v/v) (DMF, THF, DMSO or MeOH) are used. The gelation time observed using different co-solvents is included in Supplementary Table S1 as follows:

Table S1: Gelation time upon mixing GH (30 mM) with CA (90 mM) dissolved in various co-solvents.

SI No.	Solvent	Time (minutes)
1	DMF	10
2	THF	10
3	DMSO	10
4	MeOH	50

Furthermore, hydrogels prepared in PBS buffer (pH = 5.0) and co-solvent were characterised by-LC-MS. The relative abundance of different species as analysed by LC-MS is summarised in **Table S2** in the Supporting information.

Table S2: Relative abundance of different products formed from hydrazone reaction between CA and GH obtained by LC-HRMS analysis

Major product	Relative abundance in MeOH (%)	Relative abundance in DMF (%)	Relative abundance in THF (%)	Relative abundance in DMSO (%)
 C-HyG	63.4 %	87.3	90.03	-
 bis-C-HyG	29.2 %	11.9	7.8	99.3
 mono-C-HyG	7.3 %	0.8	1.2	0.3
 CA	-	-	-	0.6

In addition, the minimum gelation concentration for each co-solvent is mentioned in Supplementary **Table S3** as follows:

Table S3: MGC in different co-solvents.

Solvent	DMF	THF	DMSO	MeOH
Concentration of CA (in mM)	66	52.5	66	66
Concentration of GH (in mM)	22	17.5	22	22

In aprotic polar solvents such as DMF and THF, the relative abundance of C-HyG was found to be ~90%, with gelation occurring within 10 minutes. In contrast, DMSO yielded a weak gel within a similar

timeframe, but the relative abundance of the bis-C-HyG species was significantly higher (~99%) in DMSO. The resulting gel formed in DMSO is mechanically weaker compared to those formed in DMF or THF. Rheological measurements further support this observation, indicating that DMF and THF yield more robust and mechanically stable hydrogels. The following additions have been made to the manuscript to reflect these findings:

“Hydrazone functionalities act as effective hydrogen-bond donors and acceptors, facilitating the non-covalent interactions essential for self-assembly.^[36] Interestingly, when aqueous solutions of GH (1 eq.) and CA (3 eq.) were mixed in a 1:1 PBS buffer at pH 5.0 with a co-solvent under ambient conditions, an orange-brown hydrogel was formed. Due to the limited aqueous solubility of CA, various water-miscible organic co-solvents were tested to optimize hydrogel formation (Supplementary Fig. S4). Hydrogels were successfully obtained within 10 minutes using dimethylformamide (DMF), tetrahydrofuran (THF), and dimethyl sulfoxide (DMSO), while methanol (MeOH) required approximately one hour for gelation (Supplementary Table S1). The minimum gelation concentration (MGC) was found to be 17.5 mM in THF and 22.0 mM in DMF, DMSO, and MeOH, as determined by vial inversion tests performed after allowing the mixtures to rest overnight (Supplementary Fig. S5, Tables S3 and S4). Below the MGC, precipitation was observed instead of gel formation. The relative abundance of C-HyG in aprotic polar solvents such as DMF and THF was ~88% as assessed by LC-MS analysis (Supplementary Table S2). In contrast, DMSO yielded a weak gel, predominantly containing di-functionalized hydrazone byproduct (bis-C-HyG, ~99% abundance). This shift in product distribution is likely due to the higher density of DMSO, which restricts molecular mobility and favours the formation of dimeric structures (bis-C-HyG) over the tri-hydrazone product (C-HyG). Rheological analysis supported these observations, showing that the gel formed with DMSO exhibited the lowest mechanical strength ($G'_{\text{ax}} = 0.53$ kPa, Supplementary Fig. S6), compared to those prepared in MeOH ($G'_{\text{ax}} = 0.97$ kPa), THF ($G'_{\text{ax}} = 34.7$ kPa), and DMF ($G'_{\text{ax}} = 77.8$ kPa). These results indicate that while bis-C-HyG is capable of forming hydrogels, greater mechanical robustness is achieved when the proportion of C-HyG is higher relative to its mono- and di-substituted counterparts, as observed in gels formed using DMF and THF.”

Additionally, we would like to point out that a co-solvent ratio of 10% v/v DMSO in water was employed specifically for metal ion sensing experiments in sol state, not for gelation studies. The inclusion of DMSO was necessary to maintain C-HyG in solution under aqueous conditions, thereby preventing precipitation that could otherwise interfere with colorimetric measurements and lead to inaccurate quantification. A

detailed description of the solvent composition used in the metal ion sensing experiments is provided as follows:

“UV-Visible absorbance studies for metal ion sensing. A stock solution of purified C-HyG (10.0 mM in DMSO) was used for colorimetric sensing experiments (for solution state sensing) of metal ions. For evaluating the colorimetric sensing activity, 15 μL of the C-HyG stock solution was added to 285 μL of DMSO (10% DMSO in aqueous buffer, v/v) in a quartz cuvette (1 cm path length), followed by dilution with 2700 μL of HEPES buffer to obtain a final volume of 3.0 mL. This resulted in a final C-HyG concentration of 50 μM . Subsequently, 2 equivalents of metal salt were added, and the absorbance spectra were recorded under ambient conditions. Metal ion stock solutions (10.0 mM) were prepared in water for UV-Vis spectroscopic analysis. PBS buffer was avoided as it produced turbid solutions with certain metal ions.”

3) The description of how gels are prepared for SEM, rheology, FT-IR and other analyses lacks precise, consistent detail. In particular the solvent composition, pH, temperature, aging time and physical state (wet vs. dried, bulk gel vs. strip extract) used for each technique are not clearly stated. This makes it impossible to know whether observed differences arise from true material behaviour or simply from mismatched experimental conditions. The authors should explicitly report, for every characterization method, the exact preparation protocol and environmental parameters so that the data can be interpreted and reproduced reliably.

Response: We sincerely thank the reviewer for the careful evaluation and valuable feedback. A detailed description is included in the manuscript under the **“Experimental”** section as follows:

Gel formation. The same protocol was followed for all gelation experiments, including the determination of minimum gelation concentration (MGC), pH-dependent hydrogel formation, and the selection of a suitable co-solvent. The two building blocks, CA and GH, were dissolved separately. CA was dissolved in an organic solvent (due to its water insolubility), and GH was dissolved in buffer. Phosphate-buffered saline (PBS, 0.1 M) was used at three different pH values: 5.0, 6.0, and 7.0. The mixtures were prepared using a 1:1 ratio of buffer to co-solvent, with varying concentrations of CA and GH (see below for details). Gelation was confirmed by inverting the vials after repeated intervals of time. For all characterisation techniques which required hydrogel ageing, the gels were allowed to rest overnight and then processed further as required.

NMR. C-HyG (in purified powder form) was dissolved in DMSO- d_6 , and the 1H NMR spectrum was recorded using a Bruker FT-NMR 400 MHz instrument. Chemical shifts (δ) are reported relative to the residual solvent peak.

High resolution mass spectroscopy (HR-MS) or liquid chromatography mass spectroscopy (LCMS). High-resolution mass spectrometry (HRMS) of purified C-HyG was performed using a XEVO-G2-XS-QTOF instrument in positive ESI mode. To determine the composition of gels formed under different pH and solvent conditions, LC-HRMS analysis was also carried out using the same instrument. For analysis, hydrogels were left undisturbed overnight, then dissolved in acetonitrile to prepare 1 ppm solutions, which were subsequently analysed. To evaluate co-solvent efficacy, 30 mM GH and 90 mM CA were mixed in a 1:1 ratio of pH 5.0 PBS buffer and the respective co-solvent. For experiments involving varying precursor compositions, the concentration of GH was fixed at 30 mM (pH 5.0), while the concentration of CA was varied from 90 to 150 mM.

FT-IR spectroscopy. Gels prepared using co-solvent in PBS buffer (0.1 M) were left undisturbed overnight at ambient temperature to stabilize. The stabilized gels were then lyophilized using a lyophilizer (Alpha 1–2 LD plus, Martin Christ) to obtain powdered samples. The powdered samples were analysed using a Shimadzu FT-IR spectrophotometer (Model: IR Affinity-1) in ATR mode. Spectra were recorded in absorbance mode over the range of 4000 cm^{-1} – 400 cm^{-1} with a resolution of 2 cm^{-1} , and averaged over 32 scans.

Scanning electron microscopy (SEM). Materials prepared in PBS buffer at different pH values (5.0, 6.0, and 7.0) were lyophilized, mounted on carbon stubs, and coated with gold using an SC7620 Mini Sputter Coater/Glow Discharge System. The coated samples were then imaged using a ZEISS EVO 18 scanning electron microscope (SEM). Images were captured in secondary electron (SE) mode at magnifications of 1000 \times , 5000 \times , 10,000 \times , and 15,000 \times , using an accelerating voltage of 10 kV.

Field emission scanning electron microscopy (FESEM). To analyse the fibrous morphology, a thin film of the hydrogel was formed by mixing two precursors (30 mM GH in PBS buffer (pH= 5.0), and 90 mM CA in methanol) and depositing the mixture onto aluminium foil. After hydrogel material was formed in an hour, the foil was vacuum-dried, mounted onto a carbon stub, sputter-coated with gold using a Quorum QT 150S Plus, and examined using a Thermo Fischer FEI QUANTA 250 FEG.

Fluorescence microscopy. Immediately after mixing the two precursors (GH and CA in a 1:3 ratio; 30 mM of GH and 90 mM of CA) in a vial, a 30 μL aliquot was dropped onto a glass slide and observed under a

Leica microscope. The images captured at this point were labelled as time (t) = ~3 minutes. Thereafter, images were taken after time intervals.

Confocal laser scanning microscopy (CLSM). The precursors GH and CA were mixed in a 1:3 ratio (30 mM of GH and 90 mM of CA) in a vial, and immediately transferred to an imaging chamber. The mixture was allowed to rest for one hour to facilitate hydrogelation and was then observed using a Fluoview FV3000 (Olympus) microscope. A 488 nm laser was used to excite the system.

Rheological analysis. All rheological experiments were performed using an oscillatory rheometer (TA Instruments DHR-3 rheometer). A parallel plate geometry with a 20 mm diameter steel plate was used for all tests, with the gap height fixed at 0.5 mm. To determine the viscoelastic region of the hydrogel, freshly prepared hydrogel was placed on the lower plate, and a strain sweep was conducted in the range of 0.1–300% at a fixed angular frequency of 0.5 rad/s, whereas a frequency sweep was performed in the range of 0.1–100 rad/s under an iso-strain condition of 0.1%.

X-ray Photoelectron Spectroscopy (XPS). A 20.0 mM solution of purified C-HyG (in DMSO) was mixed with 2 equivalents of Fe(II) to obtain a precipitate, which was filtered, dried, and analysed using X-ray Photoelectron Spectroscopy (ULVAC-PHI VersaProbe 4) in elemental scan mode. A blank run of C-HyG was also recorded for comparative analysis.

pH measurements. pH of all solution or mixture was measured using a mobile Acetecq pH meter (Model: PH-035)."

The term '*hydrogel*' refers to the system used in its gel state, whereas '*powdered form*' denotes the purified or lyophilized version of C-HyG, unless otherwise specified. Additionally, the experimental procedures and conditions employed for metal ion sensing in both gel and sol states are detailed in the revised manuscript as follows:

“UV-Visible absorbance studies for metal ion sensing. A stock solution of purified C-HyG (10.0 mM in DMSO) was used for colorimetric sensing experiments (for solution state sensing) of metal ions. For evaluating the colorimetric sensing activity, 15 μ L of the C-HyG stock solution was added to 285 μ L of DMSO (10% DMSO in aqueous buffer, v/v) in a quartz cuvette (1 cm path length), followed by dilution with 2700 μ L of HEPES buffer to obtain a final volume of 3.0 mL. This resulted in a final C-HyG concentration of 50 μ M. Subsequently, 2 equivalents of metal salt were added, and the absorbance spectra were recorded under ambient conditions. Metal ion stock solutions (10.0 mM) were prepared in water for UV–Vis spectroscopic analysis. PBS buffer was avoided as it produced turbid solutions with certain metal ions.

Stoichiometric analysis. For obtaining the Job's plot, the concentrations of C-HyG (in the sol state) and Fe(II) ions were varied while maintaining a constant total molar concentration of 200 μM . The relative absorbance was plotted on the y-axis against the molar fraction of Fe(II) on the x-axis. In all cases, C-HyG was initially dissolved in DMSO and then diluted with PBS buffer to maintain a 10% DMSO content in the buffer solution. The same solvent ratio was used for determining the limit of detection (LoD) and assessing the pH stability of C-HyG in the sol state. For pH-dependent analysis, PBS buffer solutions of varying pH (1–11) and a fixed ionic strength of 0.1 M were employed.

Real-time sensing performance of C-HyG in the sol state for water samples. Three types of water samples were used for the analysis: (a) mineral water, obtained directly from commercially available brands, (b) tap water, collected from common laboratory tap of Department of Chemistry, Vellore Institute of Technology, Vellore, and (c) river water, obtained from the Palar River in Vellore, Tamil Nadu. Mineral and tap water samples were directly spiked with Fe(II) ions. In the case of river water, the sample was first decanted and filtered through a 0.2 μm syringe filter before being spiked with varying concentrations of Fe(II) ions. Subsequently, 50 μL of the C-HyG solution was added to each sample, and the absorbance was measured at 375 nm.

Real-time detection of analytes using paper strip-based sensors. Test strips were prepared by applying the mixture of CA and GH on Whatman filter paper that allowed *in situ* formation C-HyG hydrogel (30 mM) on the paper. A few drops of analyte solutions were then applied to the dried strips, followed by recording the colour change of the paper strips. Reflectance spectra for each strip (treated with either different metal ions or varying concentrations of Fe(II) ions) were measured in the range of 400–800 nm using a JASCO V-780 UV–Vis spectrophotometer. Additionally, C-HyG in the sol state (20 mM in DMSO) can also be used to prepare test strips following a similar procedure.”

4) Morphological characterization relies solely on SEM of dried gels (Figure S6). Inclusion of TEM samples would confirm fiber diameters, internal packing, and rule out drying artifacts.

Response: We thank the reviewer for the insightful comment regarding material morphology. To investigate the fibrous structure, a thin hydrogel film was prepared by mixing 30 mM GH in PBS buffer (pH 5.0) with 90 mM CA in methanol and casting the mixture onto aluminium foil. After one hour of gelation, the sample was vacuum-dried, mounted on a carbon stub, gold-sputtered (Quorum QT 150S Plus), and imaged using FESEM (Thermo Fisher FEI QUANTA 250 FEG) (Supplementary Fig. S10). The high-resolution FESEM images revealed a supramolecular nanofibrous network with an average fibre width of

$\sim 46 \pm 10$ nm, organized into rounded aggregates that retain water to form the hydrogel. Given the clear morphological features observed in FESEM, TEM analysis was not pursued further.

The following additions have been made to the manuscript to reflect these:

“Scanning electron microscope (SEM) analysis revealed that the material formed at pH = 5.0 exhibited spheroidal aggregates, which collectively arranged into a stratified architecture (Fig. 2d, and Supplementary Fig. S9a-c). Further analysis using field emission scanning electron microscopy (FESEM) enabled the visualization of an interconnecting nanofibrous network consisting of fibres with an average diameter of $\sim 46 \pm 10$ nm (Supplementary Fig. S10), indicating that these nanofibrillar networks interconnect to an intricate web forming the spheroidal aggregates. A similar spheroidal morphology was also observed at pH 6.0. However, the aggregates were significantly larger, averaging ~ 1.8 μm in diameter (Fig. 2e, and Supplementary Fig. S9d-f). These enlarged aggregates compromised the structural integrity of the hydrogel, as a weak gel was obtained. In contrast, the material formed at pH 7.0 displayed rod-like structures, with lengths of ~ 13.0 μm and widths of ~ 4.0 μm (Fig. 2f and Supplementary Fig. S9g-i). The influence of H^+ ions on hydrazone bond formation is clearly critical in determining the resulting properties of the C-HyG hydrogel. It is worth noting that under strongly acidic conditions (pH < 3), the reaction proceeds slowly, likely due to protonation of the hydrazine group, which reduces its nucleophilicity and hinders hydrazone bond formation. At pH 5.0, the addition of a colorless aqueous solution of GH (30.0 mM in PBS buffer) to a pale-yellow methanolic solution of CA (90.0 mM) at room temperature led to the formation of an orange-colored mixture, which gradually turned brown and formed a stable hydrogel (Supplementary Video V1).”

“Field Emission Scanning Electron Microscopy (FESEM) Images

Figure S7: (a-d) FESEM images of C-HyG hydrogel obtained at pH 5.0, (e) distribution curve for determining the fibre width using ImageJ.”

5) The authors are encouraged to quantify Fe(II) capture on the paper-strip sensors by measuring the decrease in Fe(II) concentration of the dipping solution before and after exposure, enabling calculation of $\mu\text{mol Fe}^{2+}$ per cm^2 of strip under identical sensing conditions.

Response: We thank the reviewer for the helpful comment regarding the practical utility of C-HyG-coated paper strips. It is worth mentioning that immersing the strips in Fe(II) solutions disrupts the gel network, making before-and-after concentration measurements impractical. Instead, applying a drop of the Fe(II) analyte onto the hydrogel-coated strip allows for effective colorimetric detection. To evaluate sensing in the gel state, 30 mM hydrogels were prepared, and applied onto paper strips, and air-dried. These strips were then tested with Fe(II) concentrations ranging from 1mM to 100 mM. A visible colour change was observed and quantified using reflectance measurements at 700 nm, which decreased progressively with increasing Fe(II) concentration (Supplementary Fig. S21).

The following inclusions have been made in the revised manuscript and revised Supporting Information.

“Therefore, CA and GH precursors were mixed and applied onto filter paper, where *in situ* hydrogelation produced disposable, ready-to-use paper-based sensors. Reflectance measurements of the hydrogel-coated paper strips showed approximately 69% reflectance (Fig. 3f). Upon exposure to aqueous solutions

of various metal salts, no significant colour change or variation in reflectance was observed (Fig. 3f and 3g). However, when Fe(II) was applied to the orange paper strip, it turned dark brown, with reflectance dropping to about 30%. Furthermore, reflectance at 700 nm decreased progressively from 80% to 9% as the Fe(II) concentration increased from 0 to 37.5 mM (Supplementary Fig. S21). At higher concentrations, no further significant changes in the reflectance spectrum were detected. This approach offers a more straightforward and practical alternative to traditional probe-doping techniques^[51] for sensor development.”

“**Real-time detection of analytes using paper strip-based sensors.** Test strips were prepared by applying the mixture of CA and GH on Whatman filter paper that allowed *in situ* formation C-HyG hydrogel (30 mM) on the paper. A few drops of analyte solutions were then applied to the dried strips, followed by recording the colour change of the paper strips. Reflectance spectra for each strip (treated with either different metal ions or varying concentrations of Fe(II) ions) were measured in the range of 400–800 nm using a JASCO V-780 UV–Vis spectrophotometer. Additionally, C-HyG in the sol state (20 mM in DMSO) can also be used to prepare test strips following a similar procedure.”

In supporting information

- **Reflectance of C-HyG coated paper strips**

Figure S21: (a) Reflectance spectrum of C-HyG coated paper strips with different concentrations of Fe(II) ions, (b) reflectance at 700 nm for concentration variance of Fe(II), (c) photographs of paper strips with different Fe(II) concentrations.

6) The manuscript does not report any ESI-MS data on the metal–ligand assemblies. Including ESI-MS and MS/MS fragmentation of extracts from both solution and gel samples would confirm the stoichiometry of Fe(II) coordination.

Response: We thank the reviewer for the valuable suggestion. High-resolution mass spectrometry confirmed the formation of a 1:1 Fe(II)–C-HyG complex, with a prominent peak at m/z 763.07 corresponding to the complex. This supports the proposed stoichiometry of Fe(II) coordination (Supplementary Fig. S17 and S18). The corresponding information has been incorporated into the revised manuscript and supplementary information as follows.

“Next, the stoichiometry of the complex formed between Fe(II) ions and C-HyG was determined using Job plot analysis,^[44,45] which revealed a 1:1 binding ratio (Fe(II):C-HyG) (Supplementary Fig. S16). Based on this result, we propose a plausible complexation model wherein Fe(II) coordinates with C-HyG through nitrogen atoms of the hydrazone linkages and oxygen atoms from the coumarin units (Supplementary Fig. S17a, b, and Video SV2). This stoichiometry was further supported by high-resolution mass spectrometry, which showed a dominant peak at m/z 763, consistent with the estimated mass of the 1:1 Fe(II)–C-HyG complex (Supplementary Fig. S18).

In supporting information

- **Plausible structure of the complex**

Figure S17: (a-b) plausible 3D structures of the optimized C-HyG-Fe(II) structure by using Chem 3D software (via MM2 energy optimization).

- **HRMS analysis of the complex**

Figure S18: HRMS spectrum of the C-HyG + Fe(II).

7) The manuscript would benefit from experiments to elucidate the coordination environment of Fe(II) within the C-HyG network and to explain its selectivity over other metal ions. A simpsuch as X-ray photoelectron spectroscopy, Mössbauer spectroscopy, or X-ray absorption spectroscopy could be applied to reveal ligand identity, oxidation state, and binding geometry and thereby substantiate the preferential binding of Fe(II).

Response: We thank the reviewer for the insightful suggestion regarding the coordination environment and selectivity of Fe(II) binding within the C-HyG network. As recommended, we conducted X-ray photoelectron spectroscopy (XPS) analysis to investigate the oxidation state and coordination behaviour of Fe in the complex. The XPS spectrum confirmed that Fe remains in the +2-oxidation state upon binding with C-HyG, indicating no oxidation occurs in the aqueous system. Additionally, a shift of approximately

0.3 eV in the N 1s binding energy, and 0.4 eV in the O 1s binding energy of C-HyG was observed upon Fe(II) coordination, suggesting that the hydrazine nitrogen is the likely binding site. These findings are further supported by FTIR and NMR analyses, which indicate a consistent binding mechanism. Accordingly, these new results have been added to the revised manuscript, and the XPS spectrum is now included in the Supporting Information (Supplementary Fig. S14).

“Additionally, X-ray photoelectron spectroscopy (XPS) revealed shifts of approximately 0.4 eV in the O 1s binding energy and 0.3 eV in the N 1s binding energy upon coordination with Fe(II), indicating the involvement of the coumarin carbonyl oxygen and hydrazone nitrogen atoms in metal binding (Supplementary Fig. S14a–d).^[38] In contrast, no significant shift was observed in the C 1s region, suggesting minimal perturbation of the carbon environment. Furthermore, the XPS spectrum displayed a characteristic Fe 2p_{3/2} peak ~709 eV and a corresponding 2p_{1/2} peak around 721 eV, characteristic of Fe(II).^[39] Notably, no signals attributable to Fe(III) were observed, confirming that iron is present in the +2 oxidation state (Supplementary Fig. S14e).”

In Supporting Information

- **X-ray Photoelectron Spectroscopy (XPS)**

Figure S14: (a) Comparative XPS spectra of C-HyG (orange line) and C-HyG with Fe(II) (brown line); (b) shift in intensity of the C 1s energy state; (c) shift in binding energy of the O 1s energy state; (d) shift in

binding energy of the N 1s energy state; and (e) Fe 2p energy spectrum in the C-HyG + Fe(II) complex, showing the Fe(II) binding energy at 209.15 eV.

8) The experiments exploring gel formation in different organic co-solvent ratios (Figure S10) and the wide panel of metal salt tests (Figure 3 d and ESI Figure S14) read as descriptive data collections without a clear link to the manuscript main hypotheses. The authors should explain why each cosolvent fraction was chosen, for example whether ten percent versus twenty percent DMSO reflects particular application environments or probes specific solvent gelator interactions, and clarify how those results advance understanding of the self-assembly mechanism. In the same way the rationale behind selecting each metal ion in the selectivity assay should be stated, for instance in terms of biological relevance or potential industrial interferents, and the results discussed to show how they support the claim of preferential Fe(II) sensing. Adding this context will convert isolated observations into a cohesive narrative that reinforces the study scientific impact.

Response: We thank the reviewer for their valuable insights regarding material formation and the role of co-solvents. In our experiments, 50% co-solvent (v/v) was employed to facilitate dissolution of the precursor CA. Reducing the co-solvent volume, it resulted in weaker gels with slower material formation. This can be attributed to the hydrophobic nature of the coumarin moieties in C-HyG, which limits efficient self-assembly in aqueous environments. However, we use 10% DMSO solution to dissolve C-HyG for metal sensing studies in sol state, followed by dilution with buffer to assess sensing performance under predominantly aqueous conditions.

Regarding the rationale behind the selection of metal ions, we considered their biological relevance (e.g., Ca²⁺, Mn²⁺), coordination potential (e.g., Zn²⁺, Cu²⁺, Ni²⁺), or environmental significance as potential interferents (e.g., Cd²⁺, Cr³⁺, Th⁴⁺). We have expanded the discussion to highlight how the minimal response from these ions, compared to the distinct signal for Fe(II), underscores the system's selectivity. This refinement provides a stronger link between experimental design and observed selectivity, enhancing the study's overall impact. The following changes have been included in the revised manuscript.

“The optical response of C-HyG was evaluated in the presence of various metal ions. No significant changes were observed upon exposure to a broad range of cations, including Ca(II), Zn(II), Ni(II), Cu(II), Co(II), Mn(II), Cd(II), Sr(II), Bi(II), Cr(III), Zr(IV), and Th(IV). These ions were selected to represent biologically abundant species such as Ca(II), and Mn(II), essential transition metals such as Zn(II), Cu(II), and Ni(II) that may compete for coordination, and environmentally relevant heavy metals such as Cd(II), Cr(III), and Th(IV) that

could act as potential interferents in sensing applications. Interestingly, Fe(II) resulted in a strong and selective response, turning the colour from yellow to dark brown (Fig. 3a). Fe(III) also led to a slight colour change to pale brown, which could be seen by eye, but it was much less noticeable than the response to Fe(II).”

“To evaluate the selectivity of C-HyG, competitive binding experiments were carried out by introducing other metal ions along with Fe(II) ion. No significant interference was observed (Fig. 3d), confirming the high selectivity of C-HyG for Fe(II).”

There are also minor comments about the figures and numerous typos present in the manuscript:

1) SEM micrographs in panels d–f require uniform, high-contrast scale bars and a slightly larger inset photograph of the vial or repositioning of the inset outside the micrograph.

Response: We appreciate the reviewer’s suggestion regarding the presentation of the SEM micrographs. We have updated the scale bars in panels d–f to ensure uniformity. Additionally, the size of the inset vial photographs has been slightly increased for better visibility. Further SEM images supporting the morphology are provided in Figure S9 of the Supporting Information.

In Supporting Information

- Scanning Electron Microscopy (SEM) Images

Figure S9: SEM images of hydrogel material obtained at (a-c) pH = 5, (d-f) pH = 6, and (g-i) pH = 7.

2) Fluorescence images in panels g–i should include scale bars and explicit labels of the time points (for example t = 0 min, t = 30 min, t = 60 min) directly on each micrograph to guide the reader through the evolution of the network.

Response: We thank the reviewer for the suggestion. In the revised manuscript, we have added scale bars to the fluorescence micrographs to enhance clarity, and time point labels have been included directly on each image to clearly illustrate the temporal evolution of the network structure.

In Supporting Information

- Fluorescence microscopy imaging

Figure S11: Fluorescence micrographs of the hydrogel (at pH = 5) captured after mixing CA and GH in stoichiometric ratio at (a) ~3 minute, (b) 30 minutes, and (c) 60 minutes (scale bar: 10 μm); and (d) change in fluorescence intensity as a function of time obtained using ImageJ.

3) Titration plots in panels c–e should include error bars or shaded confidence intervals and clearly distinguish mineral water versus tap water curves using different marker shapes.

Response: We thank the reviewer for the insightful suggestion. In the revised manuscript, we have updated the titration plots in panels c–e to include error bars representing standard deviations from three independent measurements. Additionally, we have distinguished the data sets for mineral water, river water and tap water using different marker shapes for clear visual separation.

Typo or grammatical errors:

4) Line 9–10: "its self-assembly to fibrous hydrogel material" should be replaced with "its self-assembly into fibrous hydrogel material"

5) Line 17–18: "hydrogelators, which can self-assembly through non-covalent interactions" should be replaced with "hydrogelators, which can self-assemble through non-covalent interactions"

6) Line 19–20: "numerous supramolecular hydrogel systems has been reported" should be replaced with "numerous supramolecular hydrogel systems have been reported"

7) Line 26: "Despite these challenges,spatiotemporal control" space is missing

8) Line 36: "prepared at ambient conditions form easily available building blocks" should be replaced with "prepared at ambient conditions from easily available building blocks"

9) Line 38–39: "peticularly during" should be replaced with "particularly during"

10) Line 43: "As a results, selective and sensitive detection" should be replaced with "As a result, selective and sensitive detection"

11) Line 49: "high selectivity, and straighforward readout method" should be replaced with "high selectivity, and straightforward readout method"

12) Line 75: "analyte sensor devices bacause it contains" should be replaced with "analyte sensor devices because it contains"

13) Line 77: "control in the hierarchal assembling process" should be replaced with "control in the hierarchical assembling process"

14) Line 114–116: "could not supports its own weight" should be replaced with "could not support its own weight"

15) Line 116–118: "at pH 7.0,a viscous material were produced." Space is missing

16) Line 151: "precipitation occured below CGC." should be replaced with "precipitation occurred below CGC."

17) Line 153–154: "resulting poor nucleophilicity." should be replaced with "resulting in poor nucleophilicity."

18) Line 166–167: "oscillatory rheology was employed The reaction" dot is missing

19) Line 171–173: "in 2 hours.Notably, a hydrogel" space is missing

20) Line 178: "Notable, there was no pronounced change observed" should be replaced with "Notably, there was no pronounced change observed"

21) Line 180–181: "changed form yellow to pale brown" should be replaced with "changed from yellow to pale brown"

22) Line 252–253: "The hydrogeltor C-HyG exhibits excellent sensitivity" should be replaced with "The hydrogelator C-HyG exhibits excellent sensitivity"

23) Line 255: "added advantage of its portability introduces" should be replaced with "added advantage of its portability introduces"

Response: We sincerely thank the reviewer for the thorough and insightful comments, which have significantly enhanced the quality of our manuscript. All the identified typographical and grammatical errors have been meticulously addressed in the revised version.

Reviewer #2

I have some questions and comments.

Response: We thank the reviewer for the feedback.

1) What do you mean by immobilizing water? Could you explain very briefly in the text?

Response: Thank you for the comment. Hydrogels retain substantial amounts of water within their interconnected matrix, preventing free flow while maintaining the gel's soft and hydrated properties. This process is commonly referred to as water immobilization within the gel network. We have clarified this concept by slightly modifying the sentence in the revised manuscript as follows.

“Supramolecular hydrogels are formed from low-molecular-weight (LMW) molecules, known as hydrogelators, that self-assemble via non-covalent interactions into a three-dimensional nanofibrous network capable of trapping and immobilizing water within its structure.^[1,2]”

2) Please cite after mentioning the application to avoid block references (Lines 19-21).

Response: Thank you for the suggestion. We have revised the manuscript accordingly to cite references immediately after mentioning each specific application, thereby avoiding block citations. The specific line has rewritten as follows.

“In recent years, a wide range of supramolecular hydrogel systems have been developed,^[3,4] with applications spanning petrochemicals,^[5] agricultural fertilizers,^[6] personal care products,^[7] therapeutics,^[8] and tissue engineering.^[9]”

3) Have you tested detection in a real aqueous or biological environment where sample treatment is essential? In the manuscript the only thing you mention is use in mineral water and tap water as a medium.

Response: We appreciate the reviewer’s insightful question. While our study focuses on detection performance in mineral water and tap water as representative aqueous media, we acknowledge the importance of evaluating more complex biological or environmental samples. However, real environmental waters typically contain iron predominantly in the Fe(III) oxidation state, which may necessitate pre-treatment for Fe(II) detection. To address this, we employed purified water, tap water, river water and HEPES buffer as representative media, spiking Fe(II) ions for real-time analysis to ensure accuracy. As a proof of principle, our work demonstrates that the hydrogel-based system can effectively detect Fe(II) ions in commonly used portable water sources. Towards this, we have included the following in the revised manuscript.

“To evaluate the practical applicability of C-HyG for real-time Fe(II) detection, its performance was tested in Fe(II)-spiked aqueous samples prepared using potable water sources, including mineral water, tap water, river water and HEPES buffer solution.^[47,48] UV-Vis absorption spectroscopy was used to monitor changes in absorbance at 375 nm upon the addition of Fe(II) ions at concentrations ranging from 5 μ M to 40 μ M. In the presence of aqueous C-HyG (50.0 μ M), a gradual decrease in absorbance was observed with increasing Fe(II) concentration (Fig. 3e). This trend closely resembled the results obtained under controlled buffered conditions, confirming the sensitivity of C-HyG to Fe(II) even in complex water matrices. These findings demonstrate the potential of C-HyG for Fe(II) detection in real-world water samples, with consistent sensing behaviour in the sol state.”

4) I think research should focus on the synthesis and characterization of coumarin-based hydrogelators.

Response: We thank the reviewer for the thoughtful perspective. While the synthesis and characterization of coumarin-based hydrogelators are indeed central to our study, our work demonstrates the functional application of the *in situ* prepared hydrogelator in selective Fe(II) sensing. The system has been comprehensively characterized using structural, morphological, and rheological analyses, followed by its implementation in sensing applications. We believe this integrated approach strengthens the manuscript

by bridging molecular design with practical utility. The *in situ* synthesis strategy has been emphasized in the revised manuscript as follows:

“Supramolecular hydrogels are typically prepared by altering physical or chemical conditions such as dissolving the hydrogelator at elevated temperatures, followed by cooling, applying ultrasound energy, or adjusting the pH of the solution.^[10] However, achieving controlled formation and precise spatial distribution of these materials for smart applications remains a significant challenge. This difficulty primarily arises from the weak non-covalent interactions that drive the self-assembly of LMW molecules into supramolecular structures. To overcome these limitations, *in situ* generation of hydrogelators from non-assembling building blocks offers a promising strategy for exerting spatiotemporal control over hydrogel formation.^[11] The use of catalysts to drive covalent bond formation between precursor molecules enables the controlled synthesis of hydrogelators, providing enhanced control over the hierarchical self-assembly process. This approach also improves the tunability of hydrogel properties and enhances their responsiveness to analytes in biological environments.^[12,13] In this context, the *in situ* generation of hydrazone-based LMW hydrogelators under ambient conditions, facilitated by acid or nucleophilic catalysts, represents an interesting method for the controlled fabrication of supramolecular hydrogels.^[14,15,16]”

The design principle is rewritten as follows.

“A general strategy for developing a molecular probe to sense Fe(II) ions under ambient and/or physiologically relevant conditions should consider the following criteria: (a) the probe should be readily accessible, either commercially available or easily prepared by simple mixing under ambient conditions, and must be water-soluble to ensure compatibility with aqueous environments; (b) it should contain a specific binding motif for Fe(II) ions (or other biologically relevant ions) that enables a straightforward readout, such as colorimetric detection,^[29] without the need for specialized instruments or equipment; and (c) The probe should be immobilizable on a solid support, such as a paper strip,^[30] to create a portable sensing device that allows for convenient, high-throughput detection, particularly useful in resource-limited settings.”

The hydrogelator characterisation and analysis is rewritten in the main text as follows:

“In line with the design principles for developing molecular probes capable of detecting Fe(II) ions under ambient conditions, we have designed a coumarin-based hydrogelator (C-HyG, Fig. 2a). This molecule features a coumarin moiety linked to three hydrazone groups, and is flanked by a hydrophilic guanidine

unit. Coumarin derivatives are well-known for their sensing capabilities toward anions, cations, and reactive species due to their stable optical signals, structural flexibility, and biocompatibility.^[33] However, readily accessible coumarin-based probes specifically tailored for Fe(II) detection remain scarce.^[34,35] Notably, C-HyG can be conveniently prepared *in situ* by simply mixing a solution of a coumarin aldehyde derivative (CA, Fig. 2a) with an aqueous solution of guanidine hydrazide (GH, Fig. 2a) in phosphate-buffered saline (PBS) at room temperature. The hydrogelator is formed via hydrazone bond formation, involving the reaction of one equivalent of GH with three equivalents of CA. The structure of the *in situ*-generated C-HyG has been confirmed by comparison with the isolated compound, using ¹H NMR spectroscopy, mass spectrometry, and comparative FTIR spectral analysis (Supplementary Fig. S1, S2, and S3)."

In addition to the acid-catalysed *in situ* formation of the hydrogelator and its subsequent self-assembly into hydrogel material, details on hydrogel preparation from isolated hydrogelator via the solvent-switching method have been included in the revised manuscript as follows:

"These findings demonstrate that both precursor concentration and pH can be tuned to control hydrogel formation and properties. Specifically, acid catalysis accelerates the *in situ* generation of C-HyG, promoting an interconnecting hydrogel network formation.^[Error! Bookmark not defined.] Conversely, the slower rate of hydrogelator formation at higher pH resulted in poorly structured network as weak gel or viscous material. It is worth noting that C-HyG was purified to approximately 96.2% purity, as confirmed by LC-MS analysis. However, gelation using the purified compound via a solvent-switching method was unsuccessful. C-HyG displayed limited solubility in DMSO, requiring heating to reach the desired concentration. Upon addition of buffer solutions, no hydrogelation occurred; instead, a yellow-brown precipitate formed without further network development. These findings suggest that purified C-HyG alone is insufficient for gel formation, displaying the importance of *in situ* generation of C-HyG and/or specific solvent environments in promoting supramolecular assembly."

5) What you mention is that "We believe that this *in situ* prepared hydrogel system could be beneficial for the development of readily accessible chemosensor alternatives to biomolecule-based assays (such as those using enzymes or proteins) in theranostic applications." This is not a ideal environment as you assume. In addition, what are the detection limits and linearity?

Response: Thank you for your insightful feedback. While biomolecule-based assays play a significant role in theranostics, their inherent limitations such as instability, high cost, and operational complexity highlight

the need for alternative sensing platforms. Our *in situ* synthesized hydrogel system offers a synthetic, robust, and cost-effective solution capable of encapsulating biomolecules at the point of care, making it suitable for specific sensing applications.

In terms of performance, the hydrogel achieved a Limit of Detection (LOD) of ~32 μM for Fe(II) with strong linearity ($R^2 = 0.97$), demonstrating reliable sensitivity appropriate for environmental and biomedical monitoring. While we acknowledge the need for further validation in complex biological systems, our findings highlight the hydrogel's potential as a practical and accessible alternative to traditional biomolecule-based sensors.

We have rewritten the following in the revised manuscript:

“To further examine the interaction between C-HyG and Fe(II), a UV-Vis titration experiment was conducted by gradually adding Fe(II) ions (0–125 μM) to a solution of C-HyG (50 μM) (Fig. 3c). The absorbance at 375 nm increased with Fe(II) concentration and reached saturation at ~50 μM . The observed linear relationship with the Fe(II) concentration in the range of 0-50 μM indicates that C-HyG can be potentially used for quantitatively detecting Fe(II) ions. The limit of detection (LOD)⁴⁶ was calculated to be 32.1 μM (Supplementary Fig. S19). To evaluate the selectivity of C-HyG, competitive binding experiments were carried out by introducing other metal ions along with Fe(II) ion. No significant interference was observed (Fig. 3d), confirming the high selectivity of C-HyG for Fe(II).”

Furthermore, following graph has added in the Supplementary file to support the claim.

- **Limit of detection (LoD)**

The limit of detection was calculated using the standard equation:

$$LoD = 3.3 \times \frac{\text{Standard deviation of C-HyG in sol state } (\alpha)}{\text{Slope of standard curve } (K)}$$

Where, $\alpha=0.101$, and $K=0.01037$

$$LoD = 32.14 \mu\text{M}$$

Figure S19: Plot of absorbance maxima (λ_{max}) with respect to analyte concentration, for determining the LoD using above formula.

6) In Figure 3, I observe that UV-Vis detection deviates from the proposed selectivity in the presence of other divalent cations. Specificity is not observable.

Response: We thank the reviewer for the observation. As shown in Figure 3b, a substantial decrease in absorbance at 375 nm is observed upon the addition of Fe(II) ions, while no significant change is noted for other metal ions. Figure 3d further illustrates the relative absorbance intensity (I/I_0) at 375 nm for C-HyG and various metal ions in the absence (orange columns) and presence (black columns) of Fe(II). In the presence of Fe(II), the I/I_0 ratio decreases notably below 0.7, indicating a strong interaction between C-HyG and Fe(II). In contrast, the I/I_0 values remain consistently above 0.9 for other ions, demonstrating minimal interaction. This clear distinction highlights the system's selective response toward Fe(II). The following details have been rewritten in the revised manuscript as follows:

“No notable spectral changes were detected for the majority of tested cations. However, upon the addition of Fe(II), a red shift of approximately 15 nm and a substantial decrease in absorbance at 375 nm were observed (Fig. 3b), indicating a strong interaction between C-HyG and Fe(II).”

“To evaluate the selectivity of C-HyG, competitive binding experiments were carried out by introducing other metal ions along with Fe(II) ion. No significant interference was observed (Fig. 3d), confirming the high selectivity of C-HyG for Fe(II).”

Reviewer #3:

The manuscript presents an in-situ formation of a hydrogelator and its hydrogelation due to the formation of hydrazone and the application of such a system in Fe²⁺ detection. The authors use the design of hydrazone formation between aldehyde and hydrazine, which has been shown in the past to form hydrogels of different types. However, they functionalized with a new coumarin derivative for colorimetric detection of iron salt in solution and on a paper-strip based detection platform. The work demonstrates some interesting observation on the hydrogelation properties, including pH dependence and acid-catalyzed process and selective iron salt detection capability. However, there are several claims and observations in the manuscript which are not supported by the data, as detailed below. Also, some of the experiments lack the details needed to fully appreciate the work. Therefore, I recommend a major revision.

Response: We sincerely thank the reviewer for the thoughtful evaluation and constructive feedback. We appreciate the recognition of the novelty in our approach, particularly the incorporation of a coumarin-functionalized hydrazone-based hydrogel for selective Fe(II) detection in both solution and paper-based formats. We are also encouraged by the reviewer’s interest in our findings related to acid-catalysed hydrogelation.

1) "Interestingly, when mixing aqueous solutions of CA and GH in a 1:1 ratio of hydrazide to aldehyde functional groups at pH = 5.0, under" vs " mixing precursor solutions of guanidine hydrazide (GH) and coumarin aldehyde (CA) taken in 1:3 ratio in".

I imagine that the precursors were mixed in 1:3 ratio but in the main text and SI has different ratios written which are not consistent.

Response: We thank the reviewer for the valuable insight. To ensure consistency and clarity in the presentation of gelation parameters, the necessary revisions have been made in both the main text and the Supplementary Information.

“Notably, C-HyG can be conveniently prepared *in situ* by simply mixing a solution of a coumarin aldehyde derivative (CA, Fig. 2a) with an aqueous solution of guanidine hydrazide (GH, Fig. 2a) in phosphate-buffered saline (PBS) at room temperature. The hydrogelator is formed via hydrazone bond formation, involving the reaction of one equivalent of GH with three equivalents of CA.”

Following has been added in the ‘**Experimental**’

“**Gel formation.** The same protocol was followed for all gelation experiments, including the determination of minimum gelation concentration (MGC), pH-dependent hydrogel formation, and the selection of a suitable co-solvent. The two building blocks, CA and GH, were dissolved separately. CA was dissolved in an organic solvent (due to its water insolubility), and GH was dissolved in buffer. Phosphate-buffered saline (PBS, 0.1 M) was used at three different pH values: 5.0, 6.0, and 7.0. The mixtures were prepared using a 1:1 ratio of buffer to co-solvent, with varying concentrations of CA and GH. Gelation was confirmed by inverting the vials after repeated intervals of time. For all characterisation techniques which required hydrogel ageing, the gels were allowed to rest overnight and then processed further as required.”

2) " Additionally, when the solution of CA and GH were mixed at pH 7.0, a viscous material were produced. This change can be ascribed to the increased rate of hydrazone bond formation of C-HyG due to acid catalysis, which influences the formation of supramolecular material".

This claim is not fully supported. Based on the data in Table S3, there are different ratios of product at different pH. Thus, kinetics cannot be the only factor, the chemical compositions will also contribute. Authors should consider doing the reaction in excess of CA to ensure to complete the reaction with triple substitution and then may be comparison can be made easily. In any case, authors should clarify between the chemical composition of the hydrogel vs the kinetics to influence the gel property.

Response: We thank the reviewer for the thoughtful evaluation. We have investigated the effect of using an excess of precursor CA (4–5 equivalents relative to GH) on hydrogelation. Notably, increasing the CA:GH molar ratio to 4:1 or 5:1 at pH 5.0 significantly reduces the gelation time from 60 minutes to ~10 minutes. Additionally, the yield of C-HyG in the reaction mixture increases from 63% (with 3 equivalents of CA) to around 92% (with 5 equivalents CA). These findings indicate that higher concentrations of CA enhance C-HyG formation, thereby accelerating gelation and improving gel quality. This supports a direct correlation between the *in situ* formation of C-HyG and the resulting hydrogel properties. Accordingly, the gelation kinetics can be reasonably assumed to reflect the kinetics of C-HyG formation under these conditions. Relevant additions have been incorporated into the revised manuscript as follows:

“To evaluate product formation, LC-HRMS analysis of the hydrogel at pH 5.0 showed ~63% yield of C-HyG along with 29% of a di-functionalized hydrazide byproduct (Supplementary Table S6). Notably, increasing the molar ratio of CA to GH enhanced gelation efficiency. Using 4 equivalents of CA yielded 84.4% C-HyG, while 5 equivalents increased the yield to 92.1% (Supplementary Table S7). This was accompanied by the improvement in the hydrogel's mechanical strength, with G'_{max} rising from 2.7 kPa (with 3 eq. CA) to 93.9 kPa (with 4 eq. CA), and 101.5 kPa (with 5 equivalents of CA) (Supplementary Fig. S8a). These results suggest that higher CA concentrations facilitate more efficient C-HyG formation, leading to a stronger and robust hydrogel network. Additionally, rheological tests confirmed the stability of C-HyG hydrogels, with $G' > G''$ up to 1% strain (Supplementary Fig. S8b). These findings demonstrate that both precursor concentration and pH can be tuned to control hydrogel formation and properties. Specifically, acid catalysis accelerates the *in situ* generation of C-HyG, promoting an interconnecting hydrogel network formation.^[Error! Bookmark not defined.] Conversely, the slower rate of hydrogelator formation at higher pH resulted in poorly structured network as weak gel or viscous material.”

In Supplementary Table S7

Table S7: Relative abundances of the products formed by varying concentration of CA and GH at pH = 5.0 as determined by LC-MS analysis

Product	Relative abundance GH= 30 mM CA=90 mM	Relative abundance GH= 30 mM CA=120 mM	Relative abundance GH= 30 mM CA= 150 mM
 C-HyG	63.4 %	84.4 %	92.1 %
 bis-C-HyG	29.2 %	15.6 %	-
 mono-C-HyG	7.3 %	-	7.9 %

3)" These results suggested that acid catalysis enhanced the rate of hydrogelator formation, facilitating hydrogelation by creating an interconnecting hydrogel network. Again here, why it cannot be due to the relative ratio of different product. Table S3 is not very intuitive, why the conversion of triple hydrazone product is higher in pH 7 compared to pH 5. I think, author needs to also comment on the stability of the hydrogel based on the chemical composition in different pH and not just kinetics.

Response: We thank for the critical insight on the chemical composition of the hydrogel obtained in different pH conditions. To clarify, in pH 7.0 we report a 46% of C-HyG abundance and a 50.4% of bis-C-HyG, when 3 eq. of CA reacts with 1 eq. of GH. Since the dimer is present in more concentration as compared to C-HyG, gelation is not observed for the system.

We have added following discussion in the revised manuscript:

“Overall, efficient hydrogelation depends not only by reaction kinetics but also by the selective formation of hydrazone products. While acid catalysis promotes hydrazone bond formation, the generation of C-HyG is crucial for gelation. For example, in the presence of DMSO as co-solvent, bis-C-HyG dominates at pH 5.0 (~99%, Supplementary Table S2), yielding a weak and mechanically unstable hydrogel. In contrast, increasing the CA ratio at pH 5.0 shifts the product distribution toward >90% C-HyG (Supplementary Table S7), resulting in a robust and stable hydrogel.”

4)Fig. 2 b,c, why the absorbance is in a. u.?

Response: We thank the reviewer for pointing out. The absorbance in Fig. 2b and 2c is presented in arbitrary units (a.u.) to represent relative spectral changes. However, we have now updated the figure legend with concentration of the substrates as follows:

“(b) UV-Vis spectra showing hydrazone product formation in MeOH/PBS (pH 5.0) over 60 min (green line: 0 min; blue line: 60 min) after mixing CA (90 μ M) and GH (30 μ M), (c) Conversion to hydrazone product followed by UV-Visible spectroscopy measured at 350 nm in pH 5.0 (black square), pH 6.0 (red) and pH 7.0 (blue); The conversion was followed after mixing CA (90 μ M) with GH (30 μ M)) at ambient conditions.”

5)" (e-g) Fluorescence micrographs for in situ evolution of hydrogel network at pH 5 (λ_{ex} = 490 nm)." e-g are not fluorescence micrographs. Also, they should provide the time details of the fluorescence micrographs. At what time these images were taken?

Response: We sincerely appreciate the reviewer's close attention to the detail. We have corrected the figure labels to "(g – i)" and have now labelled the acquisition time of each fluorescence image above the corresponding panel, as shown below:

Additionally, the images have been referenced in **Supplementary Figure S11**

- **Fluorescence microscopy imaging**

Figure S11: Fluorescence micrographs of the hydrogel (at pH = 5) captured after mixing CA and GH in stoichiometric ratio at (a) ~ 3 minute, (b) 30 minutes, and (c) 60 minutes (scale bar: 10 μm); and (d) change in fluorescence intensity as a function of time obtained using ImageJ.

6)Figure 2f does not look needle-like. It seems to be rolled up structure. More images can be provided in the SI.

Response: We sincerely thank the reviewer for the insight. We have incorporated following changes describing the morphology of the material obtained at pH 7, and added additional images in the supplementary Figure S9.

“In contrast, the material formed at pH 7.0 displayed rod-like structures, with lengths of $\sim 13.0\ \mu\text{m}$ and widths of $\sim 4.0\ \mu\text{m}$ (Fig. 2f and Supplementary Fig. S9g–i).”

Figure S9: SEM images of hydrogel material obtained at (a-c) pH = 5, (d-f) pH = 6, and (g-i) pH = 7.

7)"showed a general broadening of peaks, indicating that C-HyG is more specific to Fe(II) as compared to Fe(III)" This fact is not very obvious from the data in Fig. S13.

Response: Thank you for highlighting this. We would like to clarify that ^1H NMR titration experiments in DMSO-d_6 (Supplementary Fig. S15) revealed decrease in intensity of hydrazone $-\text{HC}=\text{N}-$ proton and a slight downfield shift as Fe(II) concentration increases, indicative of interaction via the hydrazone moiety. Simultaneously, we observe pronounced peak broadening, which we interpret as potential phase separation of the C-HyG-Fe(II) complex from the solution, further supporting coordination through the hydrazone nitrogen atoms; the NH group, in contrast, appears non-participatory, likely due to its less favorable electronic configuration. In contrast, Fe(III) induces aromatic proton broadening after adding Fe(III) ions in the mixture, which is consistent with its paramagnetic nature. However, we agree that specific interaction with Fe(II) as compared to Fe(III) cannot be concluded from the ^1H NMR spectroscopy.

We have included following discussion in the revised manuscript:

“Further insights into the binding mechanism were obtained via ^1H NMR titration in DMSO-d_6 . The hydrazone proton signal decreased in intensity and exhibited a slight downfield shift with increasing concentrations of Fe(II) . Additionally, peak broadening indicated the formation and possible separation of

the C-HyG-Fe(II) complex from the solution, suggesting the involvement of the –C=N– moiety in coordination (Supplementary Fig. S15). This is presumably due to coordination of the hydrazone nitrogen atoms with the metal ions, whereas the NH group does not participate, likely due to its less favorable electronic configuration for metal binding.^[40,41,42] Besides, a broadening of the aromatic proton signals was observed for Fe(III) ions, consistent with the paramagnetic nature of Fe(III) (not shown).^[43]

8)"which indicated a binding stoichiometry of 2:3 for the (Fe(II):C-HyG) complex" Here we do not know the actual concentration of the hydrazone. I imagine the concentration comes following the table S3 which shows the presence of multiple species. Thus, the 2:3 binding ratio seems like not accurate. Further, author should show if the % of various product change upon binding with FeII.

Response: We are grateful to the reviewer for this insightful observation. To clarify, while the hydrogel samples indeed consist of a mixture of products (see Supplementary Table S6), all sensing experiments and binding analyses were conducted using purified C-HyG to ensure specificity. Under these conditions, Job's plot analysis demonstrated a 1:1 stoichiometry between Fe(II) and C-HyG, indicating a direct and unambiguous binding ratio. Further validation via HRMS analysis of the Fe(II)–C-HyG complex revealed a dominant ion at $m/z \approx 763$ (Supplementary Fig. S18), which closely matches the calculated mass for the 1:1 complex, supporting our proposed coordination model (Supplementary Fig. S17). We have updated the manuscript with following information:

"Notably, purified C-HyG was used for the metal ion sensing studies. When C-HyG was prepared in situ and directly used for sensing, minor changes in the spectral response were observed compared to the use of the purified C-HyG."

"Next, the stoichiometry of the complex formed between Fe(II) ions and C-HyG was determined using Job plot analysis,^[44,45] which revealed a 1:1 binding ratio (Fe(II): C-HyG) (Supplementary Fig. S16). Based on this result, we propose a plausible complexation model wherein Fe(II) coordinates with C-HyG through nitrogen atoms of the hydrazone linkages and oxygen atoms from the coumarin units (Supplementary Fig. S17a, b, and Video SV2). This stoichiometry was further supported by high-resolution mass spectrometry, which showed a dominant peak at m/z 763, consistent with the estimated mass of the 1:1 Fe(II)–C-HyG complex (Supplementary Fig. S18)."

9)In the SI, the details of gelation experiment and the IR measurement needs to be clearer.

Response: We thank the reviewer for the suggestion. Detailed descriptions of all experimental procedures are included in the revised manuscript.

“Gel formation. The same protocol was followed for all gelation experiments, including the determination of minimum gelation concentration (MGC), pH-dependent hydrogel formation, and the selection of a suitable co-solvent. The two building blocks, CA and GH, were dissolved separately. CA was dissolved in an organic solvent (due to its water insolubility), and GH was dissolved in buffer. Phosphate-buffered saline (PBS, 0.1 M) was used at three different pH values: 5.0, 6.0, and 7.0. The mixtures were prepared using a 1:1 ratio of buffer to co-solvent, with varying concentrations of CA and GH. Gelation was confirmed by inverting the vials after repeated intervals of time. For all characterisation techniques which required hydrogel ageing, the gels were allowed to rest overnight and then processed further as required.”

“FT-IR spectroscopy. Gels prepared using co-solvent in PBS buffer (0.1 M) were left undisturbed overnight at ambient temperature to stabilize. The stabilized gels were then lyophilized using a lyophilizer (Alpha 1–2 LD plus, Martin Christ) to obtain powdered samples. The powdered samples were analysed using a Shimadzu FT-IR spectrophotometer (Model: IR Affinity-1) in ATR mode. Spectra were recorded in absorbance mode over the range of 4000 cm^{-1} – 400 cm^{-1} with a resolution of 2 cm^{-1} , and averaged over 32 scans.”

10)Fig. S5: Is this hydrogelation kinetics or kinetics for the formation of hydrazone? How was this obtained, from the UV-Vis or HPLC? Does it correspond to the formation of any hydrazone or triply functionalized derivative?

Response: Figure S5 (currently labelled as S7) illustrates the formation of C-HyG, which clearly shows an absorbance maximum at 375 nm (absorption maximum has identified from purified C-HyG). To ensure that the reaction followed second-order kinetics, we selected appropriate precursor concentrations. By monitoring the rise in absorbance at 375 nm via UV-visible spectrophotometry, we were able to determine the second-order reaction rate constants. Following discussion has incorporated in the revised manuscript, and detailed description of this procedure has provided in the Kinetic Analysis section of the supplementary file.

“The rate constant for hydrazone bond formation was determined by mixing GH (30.0 μM) in PBS buffer (pH 5.0, 6.0, or 7.0) with a methanolic solution of CA (90.0 μM), maintaining a consistent 1:1 (v/v) ratio of buffer to methanol in all experiments. Kinetic analysis, based on the increase in absorbance at 350 nm (corresponding to the absorption maximum of purified C-HyG), indicated second-order behavior, confirming the gradual formation of hydrazone bond over time.”

- **Kinetic Analysis**

The rate constant for C-HyG formation was determined by mixing GH solution (30.0 μM) in PBS buffer (pH 5.0, 6.0 or 7.0) with a methanolic solution of CA (90.0 μM), maintaining a constant 1:1 (v/v) buffer and methanol across all the experiments. The reactive groups (hydrazide from GH and aldehyde from CA) were kept at a 1:1 molar ratio to ensure second-order reaction kinetics. Product formation was monitored by measuring absorbance at 350 nm at regular intervals using a JASCO V-780 UV–Vis spectrophotometer. The second-order rate constant (k) was calculated using a second-order rate equation (Equation 1).

$$\frac{1}{[A_0]-[P]} = kt + \frac{1}{[A]} \dots\dots\dots\text{Equation 1}$$

Where, [A₀] = Initial concentration of GH, [P] = Concentration of product formed at time t minutes, and k = second order rate constant

On plotting, $\frac{1}{[A_0]-[P]}$ as the dependent variable and time (t) as independent variable, and using the linear equation (y = mx + c) in Origin 2022, the slope obtained denotes the rate constant k (Table S5). The best fits of hydrazone reaction in between CA and GH are given in Figure S7.

11)Typographical error "that can be prepared at ambient conditions form easily available building blocks"

Response: We thank the reviewer for pointing out the typographical error. We have rectified the mistake in the revised manuscript.

Revision Report for Manuscript: COMMSCHEM-25-0362A

Authors response to Reviewer/Editorial office comments:

We sincerely thank the reviewer for their time and thoughtful evaluation of our manuscript. In the revised submission, each comment has been carefully addressed. Reviewer comments are **underlined** for clarity; our responses are provided in **blue**.

Reviewer 1:

The authors have addressed the reviewers' comments thoroughly. The manuscript is technically sound, clearly presented, and suitable for publication.

Response: We sincerely thank the reviewer for their thorough evaluation and positive evaluation. We are pleased to note that the manuscript is clearly presented, and ready for publication.

Reviewer #2

Dear Authors,

I appreciate your consideration of the suggestions for improvements so that your manuscript can be published. I have read everything carefully, so I inform you that the authors of reference 15 specify a supramolecular gelling agent at an oil/water interface to produce nanofibers, not hydrazones. Could you correct this? The other references are correct.

Response: We appreciate the reviewer's careful reading and valuable feedback. Upon further review of the literature, we agree that Reference 15 does not describe hydrazone-based nanofiber formation but instead relates to supramolecular gelation at an oil/water interface. Accordingly, we have removed Reference 15 and revised the associated text. All other references have been re-evaluated and remain unchanged.

Reviewer #3:

The authors have revised the manuscript according to my comments. Therefore, I recommend its publication now.

Response: We sincerely thank the reviewer for their careful rereading of the revised manuscript. We appreciate your confidence that our revisions have satisfactorily addressed the concerns and thank you once again for your positive recommendation.

Other change(s)

The title has been updated to align with the editorial requirements as follows:

“Catalytically Controlled Formation of Coumarin-based Hydrogelator Enables Colorimetric Ferrous Ion detection in Sol and Hydrogel”